

# Generation of Rossby waves off the Cape Verde peninsula; role of the coastline

Jérôme Sirven[1], Juliette Mignot[1], and Michel Crépon[1]

[1]LOCEAN Laboratory, CNRS-IRD-Sorbonne Universités-MNHN,Paris, France

**Correspondence:** Jérôme Sirven (jerome.sirven@locean-ipsl.upmc.fr)

**Abstract.** In December 2002 and January 2003 satellite observations of Chlorophyll showed a strong coastal signal along the west african coast between $10°$ and $22°$ N. In addition, a wavelike pattern with a wavelength of about 750 kms was observed from December $20^{th}$ 2002 and was detectable for one month in the open sea, south west to the Cape Verde peninsula. Such a pattern suggests the existence of a locally generated Rossby wave which slowly propagated westward during this period. To verify this hypothesis a numerical study based on a reduced gravity shallow water model has been conducted. A wind burst, broadly extending over the region where the offshore oceanic signal is observed, is applied during 5 days. A Kelvin wave quickly develops along the northern edge of the cape, then propagates and leaves the area in a few days. Simultaneoulsly, a Rossby wave whose characterisics seem similar to the observed pattern forms and slowly propagates westward. The existence of the peninsula limits the extent of the wave to the north. The spatial extent of the wind burst determines the extent of the response and correspondingly the time scale of the phenomenon (about 100 days in the present case). When the wind burst has a large zonal and small meridional extent, the behaviour of a wave to the north of the peninsula differs from that to the south. These results are corroborated and completed by an analytical study of a linear reduced gravity model using a non-Cartesian coordinate system. This system is introduced to evaluate the potential impact of the coastline shape. The analytical computations confirm that, considering the value of the wavelength, a time scale around 100 days can be associated with the observed wave. They also show that the role of the coastline remains moderate at such time scales. On the contrary, when the period becomes shorter (smaller than 20-30 days), the behaviour of the waves is modified because of the shape of the coast. South of the peninsula, a narrow band of sea isolated from the rest of the ocean by two critical lines appears. Its meridional extent is about 100 km and Rossby waves could propagate there towards the coast.

## 1 Introduction

Eastern Boundary Upwelling Systems (EBUS) — let us quote the California, Humboldt, Canary and Benguela upwelling systems — constitute an ubiquitous feature of the coastal ocean dynamics, which has been extensively studied. They are biologically very productive thanks to a transport of nutrients from deep ocean layers to the surface, which favors the bloom of phytoplanktons. Consequently they present a strong signature, which is detectable by ocean color satellite sensors (see for example Lachkar and Gruber 2012, 2013).



The dynamics of EBUS has first been studied with conceptual models. Upwellings are created by alongshore equatorward winds (Allen, 1976, or McCreary et al., 1986) generating an offshore Ekman transport, which is compensated by a vertical transport at the coast in order to satisfy the mass conservation. The near shore pattern of the upwellings is affected by the baroclinic instability mechanism which is associated with the coastal current system and produces eddies and filaments

(Marchesiello et al., 2003). Lastly wind fluctuations modulate the upwelling intensity by generating Kelvin waves propagating poleward (Moore, 1968; Allen, 1976; Gill and Clarke, 1974; Clarke, 1977, 1983; McCreary, 1981).

At a given frequency, there is a critical latitude at which the Kelvin waves no longer exist and are replaced by Rossby waves propagating westward (Schopf et al., 1981; Clarke, 1983; McCreary and Kundu, 1985). The critical latitude decreases when the wave period shortens (Grimshaw and Allen, 1988) or when the coastline angle with the poleward direction increases (Clarke

and Shi, 1991). This latter property suggests that the shape of the coast has an impact on the upwellings.

Using a high-resolution 3D numerical model, Batteen (1997) then Marchesiello et al. (2003) confirmed that the shape of the coastline actually plays a role on the upwelling pattern. However they did not investigate by which mechanisms they are driven. In particular they did not try to confront their results with the theoretical investigations of Crépon et al. (1982, 1984) who analyzed the mechanisms responsible for this behaviour. Using an $f$-plane model, these authors showed that a cape modifies

the characteristics of the upwelling, its intensity being less on the upwind side of the cape than on the downwind side. However they did not investigate what occurs in the open sea up to 1000 km from the coast and which role could play the $\beta$ effect.

The role of the forcing has also been investigated, both from an observational and theoretical viewpoint. Enriquez and Friehe (1995) computed the wind stress and wind stress curl off the California shelf from aircraft measurements and showed, thanks to numerical experiments with a two layer model and an analytical study, that a non zero wind stress curl expands the horizontal

extent of upwelling offshore; it increases from 20-30 km to 80-100 km. The importance of the wind stress curl, which generates a strong Ekman pumping, were first emphasized by Richez et al., (1984) and later by Pickett and Paduan (2003) then Castelao and Barth (2006). They could establish that the Ekman pumping and the Ekman transport due to the alongshore winds have a comparable importance in the California Current area or off Cabo Frio in Brazil. These works did not study what occurs beyond 100 km from the coast.

In the present paper, we study the role of the wind stress and of the coastline geometry in generating mesoscale anomalies offshore, up to a distance of 500-1000 km off the coast. Both a numerical and an analytical point of view are adopted. The departure point is the observation of a wave-like pattern on Chlorophyll satellite observations off the Senagalese coast, in the region of the Sénégalo-Mauritanian upwelling (see Lathuillière et al., 2008, and Farikou et al., 2015 to find further information about the Chl-a variability and the upwellings off the west African coast, Capet et al., 2017 for a recent analysis of the small

scale variability close to the Senegal and Gambia coasts, and Kounta et al., 2018 for a detailed study of the slope currents along west Africa). Attention is focused on offshore mesoscale activity associated with the upwelling, a recurrent feature of upwelling systems (see Capet et al., 2008 a, b).

The paper is articulated as follows. Observations of Chlorophyll in 2002-2003 off the west African coast are shown and described in section 2. In section 3, a numerical study with a on linear reduced gravity shallow water model with a single active





layer on the sphere is conducted; it shows that a wind stress anomaly active during a few days can generate a pattern that seems very similar to the observed one. The impacts of the wind anomaly extent and of the coastline geometry are also briefly studied. In section 4, a theoretical analysis of the wave dynamics in the vicinity of a cape is conducted to confirm and enlarge upon the results obtained in the previous section, using a linear shallow water model in a non-Cartesian coordinate system.

## 5  2  Observation of a wave off the Cape Verde peninsula from an ocean color satellite sensor.

The Senegalo-Mauritanian upwelling off the west coast of Africa forms the southern part of the Canary upwelling system. This region has been intensively studied by analysis of SeaWiFS ocean-color data and AVHRR sea-surface temperature as reported in Demarcq and Faure (2000), and more recently by Sawadogo et al. (2009), Farikou et al. (2013, 2015), Ndoye et al. (2014), and Capet et al. (2017). These studies indicate that the presence of an intense upwelling is attested by both ocean-color and

sea-surface temperature signals. Moreover this upwelling shows a strong seasonal modulation. It starts to intensify in October, reaches its maximum in April and slows down in June. Very high chlorophyll-a concentration are observed near the coast when the maximum is reached. However the concentration rapidly decreases offshore (Farikou et al., 2013, 2015; Sawadogo et al., 2009), suggesting that the upwelling extent and the eddy activity in this region is less than in other upwelling systems like the Californian Upwelling system (Marchesiello and Estrade, 2009, Capet et al., 2017).

From December $20^{th}$ 2002 up to January $8^{th}$ 2003, a strong chlorophyll signal was observed along the African coast on SeaWIFs satellite images, between $10°$N and $22°$N, indicating an intense biological activity. In addition, a well-defined "sine-like" pattern (Figure 1) was observed on ten images taken at ten non-consecutive different days, which excludes a possible artifact due to image processing. This pattern is located between $12°$N and $14°$N east of $20°$W and extends offshore up to $22°$W in a region which is off the coastal upwelling zone. West of $20°$W the signal seems to have a larger meridional extent,

reaching $16°$-$17°$N (see panel 3).

This pattern, which broadly keeps the same form during a twenty day interval, slowly progresses westward at a speed not exceeding 5 cm s$^{-1}$ (a more precise estimation of the speed from the observations is quite hazardous). After January $8^{th}$ 2003, a cloudy period of several days occurred, which prevented satellite observations. At the end of this episode (January $15^{th}$) the "sine" pattern was no longer visible. This episode might be the signature of a Rossby wave propagating westward.

As this phenomenon lasts at least one month, its typical time-scale is expected to range between one and a few months.

An enhanced chlorophyll concentration, as seen by the satellite sensor, is the signature of growing phytoplankton. A local development of phytoplankton benefits from an increase of the nutrient concentration in the surface layers of the ocean. The latter may be generated by an enhanced mixing in the surface layers associated with an increased turbulence due to the Kelvin or Rossby wave activity and by the vertical velocity associated with the divergent Rossby waves. Another possible process

explaining this growth could be the advection linked with Rossby waves – they create a current anomaly which can transport nutrients and phytoplankton from the upwelling area where they are highly concentrated. However, this mechanism is slow;





for a current anomaly of 5 cm s$^{-1}$, the transport of a parcel of fluid over 500 km – less than the zonal extent of the offshore pattern – would need about one year.

The signal along the coast is also modulated by coastal Kelvin waves propagating northwards. Clarke and Shi (1991) showed that they can propagate when their angular frequency is larger than a critical angular frequency $\omega_c = c\beta\cos\Theta/2f_0$. In this formula $\Theta$ is an angle taking into account the tilt of the coastline with a meridian, $c$ a typical velocity of a baroclinic mode and $f_0$ and $\beta$ the usual parameters linked to the Earth's rotation. These authors found that $\omega_c$ ranges between 84.69 and 115.2 days around the Cape Verde peninsula (see their table 2a). These values are close to the characteristic time scales we expect here.

The existence of a long period divergent baroclinic Rossby wave could explain the offshore sine-like pattern described here. To investigate this hypothesis, we first present numerical experiments made with a numerical shallow water model. They help us see how such a wave is created and elucidate its nature. Then, a theoretical analysis is presented in order to understand how the Kelvin and Rossby waves behave when the coast presents a cape and to complete the interpretation of the numerical experiments. A wide range of periods is explored, going from 10 days to one year.

## 3 Numerical study

### 3.1 The model

The numerical model is a reduced gravity model on the sphere with one active layer of thickness $h$. It extends over an infinite layer at rest. The velocity $\mathbf{v}$ in the active layer and the thickness $h$ verify the equations

$$\partial_t h + \operatorname{div}(h\mathbf{v}) = 0 \tag{1}$$

and

$$\partial_t \mathbf{v} + (\operatorname{rot}\mathbf{v} + f)\mathbf{n}\times\mathbf{v} = -\mathbf{grad}\Phi + \frac{\boldsymbol{\tau}_0}{h} - \frac{r}{h}\mathbf{v} + \nu\Delta_H\mathbf{v} \tag{2}$$

where $\mathbf{n}$ is a vector normal to the Earth's surface and $\operatorname{rot}\mathbf{v} = (\nabla\times\mathbf{v}).\mathbf{n}$. The function $\Phi$ is equal to $g^\star h + \mathbf{v}^2/2$ where $g^\star$ is the reduced gravity.

We assume for simplicity that the vector $\boldsymbol{\tau}_0$, which represents the surface wind stress divided by the ocean density, derives from a potential $\phi_0(x,y)$: $\boldsymbol{\tau}_0 = -\mathbf{grad}\phi_0$ (implying an irrotational mean wind). This hypothesis allows us to compute explicitly the obtained mean state. Indeed $\mathbf{v} = \mathbf{0}$ and $g^\star h_0^2/2 = -\phi_0 + C_0$ is an obvious solution of the previous system (the constant $C_0$ is determined by using the fact that the mean value of $h_0$ remains unchanged during the integration). It will also facilitate the analytical computations made in the next section. A more complex set-up could be used for the numerical experiments but it will be seen below that this one suffices.





## 3.2 Numerical resolution

The model domain is closed and centered at 15°N, the latitude of the Cape Verde peninsula; it has a latitudinal extent of 20°
and a longitudinal extent of 30°. The peninsula is modeled as indicated in Fig. 2 in order to mimic the geometry of the coast
of Senegal (simplified and smoothed). The mean value of $h_0$ is equal to 200 m and the reduced gravity $g^\star$ to 0.02 m s$^{-2}$.
Consequently the Rossby radius of deformation $R_0 = \sqrt{g^\star h_0}/f_0$ at the latitude of the Cape Verde is equal to 53 km.

The previous equations are solved by finite differences on a C-mesh on the sphere, the mesh size being equal to $(1/12)°$ in
longitudinal and latitudinal directions. The spatial scheme preserves enstrophy, following Sadourny (1975). No slip boundary
conditions are applied. There is no added dissipation in the continuity equations and mass is conserved by the numerical
scheme. The time integration is performed using a leapfrog scheme with a time step of 300 seconds. The viscosity $\nu$ and the
coefficient $r$ of equation (2) are respectively equal to 28 m$^2$s$^{-1}$ and $8 \times 10^{-5}$ m s$^{-1}$ ($h/r \simeq 1$ month).

## 3.3 Numerical results

A constant mean wind stress of amplitude equal to 0.06 N m$^{-2}$ (corresponding to a mean wind velocity of about 5 m s$^{-1}$ and a
value of $\boldsymbol{\tau}_0$ equal to $6 \times 10^{-5}$ m$^2$ s$^{-2}$) and oriented along a south-south-west direction is applied during four years from a rest
state. A stationary mean state, which verifies the theoretical relation given in section 3.1, is at that time reached. A north-south
wind stress anomaly which extends over approximately 500 km and whose maximum is still equal to 0.06 N m$^{-2}$ is then
applied during five days (see Fig. 2 first panel). The integration is continued during 45 days, after the anomaly has disappeared.

After the wind stress anomaly has vanished, the subsequent states of the ocean are shown every five days in Fig. 2 in terms
of the active layer thickness. A coastal Kelvin wave forms north of the cape and quickly propagates along the coast. After 5
days, it has already gone beyond 25° N and after 10 days only the remains of the wave are still visible. South of the cape,
a well marked wave pattern develops. Its size matches more or less the size of the wind stress anomaly. It slowly propagates
westwards with a velocity of about 3-4 cm s$^{-1}$. The amplitude of the wave decreases quickly because of the large value of $h/r$.
The minimum value of $h$ is about $-3.5$ m when the wind ceases (second panel) and reaches only $-2$ m after 25 days (seventh
panel).

These characteristics are those of a Rossby wave locally generated by a wind anomaly, then freely propagating in the open
ocean. The wavelength of this wave is about 750 km ($k_R \simeq 0.84 \times 10^{-5}$ m$^{-1}$). Such a value is compatible with the theoretical
study presented in the next section when the period of the wave is about 100 days. This modeled response to a wind stress
anomaly also matches the satellite observed signal described in the previous section. It thus suggests that the latter is the
consequence of the existence of Rossby wave generated by a wind burst.

Though the duration of the wind burst is short in our numerical experiment (5 days), the response of the system privileges
a much longer time scale, exceeding 2 months. This result is not inconsistent. Indeed the Fourier transform of a rectangular
pulse is the sine-cardinal function. It thus contains a significant amount of energy at low frequencies and thus can generate a
low frequency response as the one observed here.





Figure 3 shows the results obtained in a comparable experiment when the cape is absent. The response of the model is very similar: in the area of the wind anomaly, a Kelvin wave forms then quickly desappears whereas a more persistent Rossby wave slowly propagates westward. However the meridional extent of the Rossby wave is broader east of 18° W, in the area where previously was the cape. This suggests that the cape simply limits the extent of the wave northward but does not modify its

dynamics. The theoretical study of section 4 will confirm that the role of the cape remains moderate at low frequency.

A question arises: why does the wave have such a wavelength ? Numerical experiments clearly show that the longitudinal wavelength is defined by the spatial scale of the forcing anomaly. This is first illustrated in Figure 4, which shows the response of the model to a wind burst whose extent is four times smaller than the initial wind burst of Figure 2. The latitudinal and longitudinal extent of the model response are approximately divided by two as expected. No Kelvin wave of significant amplitude

is generated because there is no longer wind anomalies at the coast; indeed, the centre of the wind anomaly is unchanged in comparison with the reference experiment. We will show in section 4 that the period associated with the wave are increased when the wavelength is reduced (reaching approximately 150 days).

Two supplementary experiments (Figures 5 and 6) have been made, in which a wind burst of large longitudinal (about 1000 km) and small latitudinal (about 100 km) extent is applied during 5 days; the anomaly is centered at 14°N in one case and

17°N in the other (see the first panel in Figures 5 and 6). These anomalies create a Rossby wave with a large zonal extent. However the response of the model differs in the two cases. When the anomaly is located south of the cape, a Kelvin wave is generated and a negative anomaly appears between 26° and 20°W. However no positive anomaly with a comparable amplitude can be seen closer to the coast. A weak signal appears after day 20 but its extent is very small and its amplitude is four times smaller than the amplitude of the anomaly observed around 25°-27°W. South of 14°, a wave of small amplitude is created and

propagates southward. Its latitudinal wavelength is comparable with the latitudinal extent of the wind anomaly. When the wind anomaly is located north of the cape, a negative anomaly appears between 26° and 20°W as previously; besides, a positive anomaly can be seen from day 5 and its amplitude is half the amplitude of the signal observed around 25°-27°W.

Clearly the response of the model close to the cape depends on the location of the wind anomaly, north or south of the cape. When the latter acts south of the cape, the anomaly which forms around 15°W is small and quickly desappears; a wave

which propagates southward seems to prevent its existence. On the contrary, when the wind anomaly acts north of the cape, an anomaly forms around 15°W and gets stuck in this place; the wave which propagates southward still exists but its amplitude is about twice samller than in the previous case. In the next section, we analytically investigate this dissymmetric behaviour in an idealized case.

## 4   Analytical study

In this section we generalize the approach followed by Clarke and Shi (1991) in order to study the response of the ocean at distances several times longer than the size of the Cape Verde peninsula (i.e. comparable with the sine pattern observed in the





satellite data). We thus suppose that the eastern coast presents a cape and analyze how the free waves are influenced by its existence.

A new system of coordinates is introduced over the entire domain. It allows us to establish equations valid over the whole basin but with a very simple boundary condition on the domain frontier where the cape is located. By using the WKB approx-

imation and making the hypothesis that the mass transport parallel to the coast is much larger than the perpendicular one, we can investigate the impact of the curvature of the coastline on the Rossby wave dynamics.

## 4.1 A model for the Kelvin and Rossby waves

We consider a reduced gravity shallow water model in the $\beta$ plane forced by a constant wind stress which derives from a potential $\phi_0(x,y)$ ($\boldsymbol{\tau}_0 = -\mathbf{grad}\phi_0$), as in section 3. Numerical integrations have shown that the exact solution $\mathbf{v} = \mathbf{0}$ and

$g^\star h_0^2/2 = -\phi_0 + C_0$ is actually obtained after a few year integration (see section 3.1 and 3.3).

If an anomaly $(\tau_x, \tau_y)$ is added to the mean forcing $\boldsymbol{\tau}_0$, a perturbation is generated; it is characterized by a *depth anomaly h* (the thickness of the first layer is now $h_0 + h$) and a velocity $\mathbf{v}$. A linear approximation is sufficient to study the first steps of the evolution of the perturbation if the forcing anomaly remains moderate. The anomaly $(\tau_x, \tau_y)$ can be written in terms of a potential $\phi(x,y,t)$ and a stream function $\psi(x,y,t)$,

$$
\begin{aligned}
\tau_x &= -\partial_x\phi - \partial_y\psi \\
\tau_y &= -\partial_y\phi + \partial_x\psi
\end{aligned}
\tag{3}
$$

so that the divergent part of the forcing is given by $-\Delta\phi$ and the rotationnal part by $\Delta\psi$.

The equations verified by *the anomalies h* and $\mathbf{v}$ are thus:

$$
\begin{aligned}
h_0\partial_t u - fh_0 v + \partial_x(g^\star h_0 h + \phi) &= \nu h_0 \Delta_H u - ru - \partial_y\psi \\
h_0\partial_t v + fh_0 u + \partial_y(g^\star h_0 h + \phi) &= \nu h_0 \Delta_H v - rv + \partial_x\psi \\
\partial_t h + \partial_x(h_0 u) + \partial_y(h_0 v) &= 0
\end{aligned}
\tag{4}
$$

The role of the diffusion and dissipation will not be considered below – a smoothing and damping of the solution is expected

when it is taken into account. The previous system may be further simplified by introducing the zonal and meridional transports $T_x = h_0 u$ and $T_y = h_0 v$ and a potential $\eta$ equal to $g^\star h_0 h + \phi$. It becomes

$$
\begin{aligned}
\partial_t T_x - fT_y + \partial_x\eta &= -\partial_y\psi \\
\partial_t T_y + fT_x + \partial_y\eta &= \partial_x\psi \\
\partial_t\eta + c^2[\partial_x T_x + \partial_y T_y] &= \partial_t\phi
\end{aligned}
\tag{5}
$$

where $c = \sqrt{g^\star h_0}$ is a function of $x$ and $y$.

These equations apply inside the ocean domain, whatever its shape. A boundary condition is added along the domain frontier

(the normal transport vanishes) but the latter is difficult to handle when the shape of the coast is complex. Lastly, the spatial mean value of the depth anomaly $h$ remains null.



The propagation of a Kelvin wave along the eastern boundary and the possible generation of Rossby waves can be studied by using system (5). Here, we consider an eastern boundary whose angle with the meridians smoothly varies and we seek to understand how these variations may affect the wave dynamics. We consider only the case of a cape even though the method could be applied in other cases. A coordinate change is made in order to "straighten the coast" and therefore have a simple boundary condition; equations (5) are correspondingly modified to match the new coordinates.

A new orthogonal system of coordinates $X = \mathcal{X}(x,y)$ and $Y = \mathcal{Y}(x,y)$ is thus introduced such that the eastern boundary is now defined by the simple equation $X = 0$ (rather than a complex one such as $f(x,y) = 0$). In the local orthonormal basis $\mathbf{e}_X, \mathbf{e}_Y$ associated with this coordinates system, the line element $\mathrm{d}\mathbf{l}$ reads

$$\mathrm{d}\mathbf{l} = a\,\mathrm{d}X\,\mathbf{e}_X + b\,\mathrm{d}Y\,\mathbf{e}_Y$$

where $a$ and $b$ are geometrical factors which convey the stretching of the coordinates along orthogonal directions (note that the relations

$$a = \sqrt{(\partial_X x)^2 + (\partial_X y)^2} \quad \text{and} \quad b = \sqrt{(\partial_Y x)^2 + (\partial_Y y)^2}$$

where the initial coordinates $x$ and $y$ are now functions of the new coordinates $X$ and $Y$ are used to compute $a$ and $b$).

Such a coordinate change is illustrated in figure 7 for a cape protruding into the sea over a distance of 80 km. Only half of the symmetric domain is shown. The initial coordinates $x$ and $y$ are the zonal and meridional coordinates; the new coordinates $X$ and $Y$ are represented in the original system and some of their values are indicated. Other coordinate changes would be possible. The corresponding geometrical factors $a$ and $b$ are shown in figure 8. They differ from 1 respectively in a close neighbourhood and to the west of the cape ($a$ is equal to 0.1 around the extremity of the cape whereas $b$ reaches 40 at 300 km west of the cape). A detailed study of this example will be presented in subsection 4.3.

Using the coordinates $(X, Y)$, system (5) becomes

$$
\begin{aligned}
\partial_t T_X - F T_Y + a^{-1}\partial_X \eta &= -b^{-1}\partial_Y \psi \\
\partial_t T_Y + F T_X + b^{-1}\partial_Y \eta &= a^{-1}\partial_X \psi \\
\partial_t \eta + (C^2/(ab))[\partial_X(bT_X) + \partial_Y(aT_Y)] &= \partial_t \phi
\end{aligned}
\tag{6}
$$

where $F$, $C$, $T_X$, $T_Y$ are functions of $X$ and $Y$ corresponding to $f$, $c$, $T_x$ and $T_y$ (the notations for $\eta$, $\phi$ and $\psi$ have not been changed to enhance the readability, but these functions also depend on $X$ and $Y$).

We now concentrate on processes whose time scale is much larger than one day. Moreover we consider only the evolution of free waves. This situation corresponds to the case numerically investigated in section 3 and illustrated in Figures (2 - 6): the wind stress anomaly that had created the depth anomalies has ceased to exert. With these hypotheses, system (6) can be reduced into the following equation which characterizes the evolution of $\eta$:

$$
\partial_t \left[ \eta - R_0^2 \left( \frac{1}{a^2}\partial_{XX}^2 \eta + \frac{1}{b^2}\partial_{YY}^2 \eta \right) \right] - \cdots
$$
$$
\cdots \frac{R_0^2}{ab} \left[ F^2 \partial_X \left( \frac{b}{aF^2} \right) \partial_{tX}^2 \eta + \partial_Y F \, \partial_X \eta + F^2 \partial_Y \left( \frac{a}{bF^2} \right) \partial_{tY}^2 \eta - \partial_X F \, \partial_Y \eta \right] = 0
\tag{7}
$$



where $R_0^2 = C^2/F^2$ is the Rossby radius. The boundary condition at the eastern coast now reads:

$$\text{at} \quad X = 0, \quad b\partial_{tX}^2\eta + aF\partial_Y\eta = 0 \tag{8}$$

for all $t > 0$ and $Y$. The details of the computations can be found in Appendix 1.

When $a$ and $b$ are close to 1 (see Figure 8) the new coordinates system is nearly similar to the original one; indeed $F$ no

longer depends on $X$ but only on $Y \simeq y$, and in those regions where this condition is verified equation (7) simplifies:

$$\partial_t \left[\eta - R_0^2(\partial_{XX}^2\eta + \partial_{YY}^2\eta)\right] - R_0^2 \left[\partial_Y F \, \partial_X\eta + F^2\partial_Y(\frac{1}{F^2})\partial_{tY}^2\eta\right] = 0 \tag{9}$$

We recognize the equation characterizing the propagation of waves in the $\beta$ plane ($\beta = \partial_Y f$) for a shallow water model. For spatial scales much larger than $R_0$ and at low frequency, equation (9) can be further simplified. It becomes $\partial_t\eta - R_0^2\partial_Y F \, \partial_X\eta = 0$ which simply models the westward propagation of long Rossby waves.

The coefficient $b$ decreases as one gets closer to the cape (see Figure 8). An impact of the cape in the open sea may therefore be expected because of the terms proportional to $\partial_X(\frac{b}{aF^2})\partial_{tX}^2\eta$ and $\partial_X F \partial_Y\eta$ in equation (7).

### 4.2   Ray theory

When a wave propagates in a medium whose properties spatially change, it does not follow straight lines but more complex paths. Ray theory – or WKBJ approximation – is used to determine the paths followed by the waves in such a medium. It

applies when the wavelength is smaller than the typical scale at which the properties of the medium vary. The spatial variations of the components $k$ and $l$ of the wavevectors are taken into account by introducing a complex function $\theta(X,Y) = \theta_R + i\theta_I$ such as $k = \partial_X\theta$ and $l = \partial_Y\theta$. The potential $\eta$ is then equal to

$$\eta = \eta_0(X,Y)\exp[i(\omega t + \theta(X,Y))] \tag{10}$$

where it is assumed that $|\eta_0^{-1}\partial_X\eta_0| \ll |\partial_X\theta|$, $|k^{-1}\partial_X k| \ll |\partial_X\theta|$, $|\eta_0^{-1}\partial_Y\eta_0| \ll |\partial_Y\theta|$, $|l^{-1}\partial_l l| \ll |\partial_Y\theta|$. As required by the

theory, these inequalities ensure that the wavelength is smaller than the typical scale of variation of the system, here conveyed by $R_0$, $F$ and the coefficients $a$ and $b$. Excepted in a very close vicinity of the cape (distance smaller than around ten kilometers) and for the meridional wavelength in the area located between -50 and 50 km north and south of the cape and beyond 200 km west of the cape, these inequalities mean that several wave patterns must be visible in the considered domain. This condition is verified for the waves observed in the numerical experiments and for the waves considered below.

Considering these hypotheses, $\eta$ is given from equation (10) in the neighbourhood of a point $M_0$ of coordinates $(X_0, Y_0)$ by the approximate expression

$$\eta(X,Y) = \eta_0(X_0,Y_0)\exp[i(\omega t + k(X - X_0) + l(Y - Y_0))]$$

where $k = \partial_X\theta|_{M_0}$ and $l = \partial_Y\theta|_{M_0}$. The physical meaning of this solution is explicited by taking its real part:

$$\Re(\eta) = \eta_0(X_0,Y_0)e^{-k_I(X-X_0)-l_I(Y-Y_0)}\cos[\omega t + k_R(X - X_0) + l_R(Y - Y_0)]$$



The solution must not increase westward since it cannot become infinite, which implies $k_I < 0$. Note however that $k_I > 0$ is possible if this occurs only in a (small) bounded domain. For $k_R > 0$, the wave propagates westwards.

The values of $k$ and $l$ are obtained by computing the function $\theta$ from equations (7) and (8). After simplifying them by using the hypotheses, we find that $\theta$ verifies the approximate (eikonal or Hamilton-Jacobi) equation

$$\left(1 + R_0^2\left[\frac{1}{a^2}(\partial_X\theta)^2 + \frac{1}{b^2}(\partial_Y\theta)^2\right]\right) - \cdots$$

$$\cdots \frac{R_0^2}{ab}\left[iF^2\partial_X(\frac{b}{aF^2})\partial_X\theta + iF^2\partial_Y(\frac{a}{bF^2})\partial_Y\theta + \partial_Y(\frac{F}{\omega})\partial_X\theta - \partial_X(\frac{F}{\omega})\partial_Y\theta\right] = 0 \tag{11}$$

with the boundary condition

$$ib\partial_X\theta + a(F/\omega)\partial_Y\theta = 0 \tag{12}$$

at $X = 0$.

To make the computations clearer we set

$$- z_1 = (\partial_X\theta)R_0/a = kR_0/a$$
$$- z_2 = (\partial_Y\theta)R_0/b = lR_0/b,$$
$$- w_1 = (R_0/2b)\left[\partial_Y(\frac{F}{\omega}) + iF^2\partial_X(\frac{b}{aF^2})\right]$$
$$- w_2 = (R_0/2a)\left[-\partial_X(\frac{F}{\omega}) + iF^2\partial_Y(\frac{a}{bF^2})\right]$$

and the problem (11) associated with the boundary condition (12) is rewritten as the following system :

$$\partial_Y(\frac{az_1}{R_0}) = \partial_X(\frac{bz_2}{R_0}) \tag{13}$$

$$1 + z_1^2 + z_2^2 - 2z_1w_1 - 2z_2w_2 = 0 \tag{14}$$

$$z_2 = -i\frac{\omega}{F}z_1 \quad \text{at} \quad X = 0 \tag{15}$$

At the boundary, equations (14) and (15) permit to determine the values of $z_1$ and $z_2$, hence of $k$ and $l$. Indeed, the introduction of (15) in (14) leads to the equation

$$1 + (1 - \frac{\omega^2}{F^2})z_1^2 - 2z_1(w_1 - i\frac{\omega}{F}w_2) = 0$$

Setting $\xi = \sqrt{1 - (\omega/F)^2}$, this equation can be rewritten

$$J(\xi z_1) = \frac{1}{\xi}(w_1 - i\frac{\omega}{F}w_2) = W_R + iW_I \tag{16}$$

where $J(z) = \frac{1}{2}(z + \frac{1}{z})$ is the Joukowsky transform of $z$; $W_R$ and $W_I$ are given by the relations

$$W_R = \frac{R_0F}{2\xi} \times \frac{F}{\omega}\left[-\xi^2\frac{1}{b}\partial_Y\frac{1}{F} + (1-\xi^2)\frac{1}{a}\partial_Y(\frac{a}{bF})\right]$$
$$W_I = \frac{R_0F}{2\xi} \times \frac{1}{b}\partial_X(\frac{b}{aF})$$





Equation (16) is the dispersion relation of the wave at the boundary. The Joukowsky transform of $\xi z_1$ (or equivalently of $i\dfrac{F}{\omega}\xi z_2$) depends on the frequency $\omega$, the mean state (through $R_0$), the latitude (through $F$) and the geometry of the coast (through $a$ and $b$).

The sketch in Figure 9, which shows the half complex plane $\Im(z) < 0$ (upper panel) and the complex plane $W = J(z)$ (lower panel), explicits how the Joukowsky transformation works. The inferior half plane has been chosen since we expect $k_I < 0$, or in other words $\Im(z) < 0$.

— When $W_I = \Im(W)$ is *positive*, the complex number $\xi z_1$ such as $J(\xi z_1) = W = W_R + i\,W_I$ has a norm smaller than 1 (it corresponds to the gray areas in Figure 9). The wavelength is thus large ($\lambda > 2\pi \xi R_0 /a$). If $W_R = \Re(W)$ is positive, the phase speed is negative (westward propagation).

— When $W_I = \Im(W)$ is *negative*, the complex number $\xi z_1$ such as $J(\xi z_1) = W = W_R + i\,W_i$ has a norm larger than 1 (it corresponds to the white areas). The wavelength is thus small. If $W_R = \Re(W)$ is positive, the phase speed is still negative.

— When $W_I$ *vanishes*, a situation which occurs when the coast is a straightline, the unique complex solution previously found can cease to exist. Indeed, if $|W_R|$ is smaller than 1, there is *one complex* solution whose norm is equal to 1 (on the half circle in bold in Figure 9); and if $|W_R|$ is larger than 1, *two real* solutions are obtained (between $]-1, 1[$ and outside this interval). The case ($W_I = 0$) is detailed in the next subsection.

At the coast, relation (15) determines $z_2$ when $z_1$ is known. It implies that

$$l_I = \frac{\omega}{F}\frac{b}{a}k_R \quad \text{and} \quad l_R = -\frac{\omega}{F}\frac{b}{a}k_I$$

For low frequencies $\omega/F$ is much smaller than 1 and $l_I$ and $l_R$ can be ignored. An approximate expression of the waves is

$$\eta(X,Y,t) = \eta_0(X_0,Y_0)e^{-k_I(X-X_0)}\cos(\omega t + k_R(X-X_0))$$

and the dynamics is controlled by the westward propagation of Rossby waves. For shorter periods, the situation may be more complex because the ratio $\omega b/(aF)$ may be close to 1. This effect is in agreement with the results obtained in section 3, where a southward propagation of waves was observed in the open ocean, simultaneously with the westward progation of Rossby waves.

Knowing $z_1$ and $z_2$ along the coast, it is possible to continue the resolution of the problem (13-14) and compute explicitely the rays characterizing the propagation of the waves. An algorithm which fulfils this objective is described in appendix 2. However, under the supplementary assumption that $|T_X|$ remains much smaller than $|T_Y|$ up to a distance from the coast of about a few Rossby radius of deformation, approximate expressions for $z_1$ and $z_2$ can be obtained analytically. This also permits to initialize the algorithm described in appendix 2.

### 4.3 Analytical study for $|T_X| \ll |T_Y|$.

In this case, equation (15) which is exact at the coast $X = 0$ can also be used on a band of a few hundred kilometers off the coast and yields a good approximation of the solution (for a more detailed discussion, see Appendix 2). Consequently, equation





(16) becomes valid over a domain which extends far off the coast in the open ocean. In this section, the consequences of this relation are briefly presented for a straight coastline, then investigated in detail for the cape shown in figure (7). In agreement with the hypotheses of the previous subsection, we assume that $\omega \ll F$ but the results and graphics will be produced up to the limit value $\omega = F$.

**Case of a straight coastline**. This case has been extensively studied in the literature — let us mention Richez et al. (1984), Grimshaw and Allen (1988), Clarke and Shi (1991), McCalpin (1995), Liu et al. (1998) — each article stressing a particular issue. Using the previously established equations, we summarize here known results about the existence of critical frequencies along an ocean boundary.

For a straight coastline following the south north direction, we have $X = x$ and $Y = y$; consequently $a = b = 1$ and $F$
depends only on $Y$. Thus $W_I = 0$ and

$$2W_R = \frac{R_0\beta}{\omega}\frac{2\xi^2 - 1}{\xi}$$

where $\beta = \partial_Y F$.

In figure 10 (top panel), the coefficient $W_R$ is shown at different latitudes as a function of the period. At 15° N, the critical value $W_R = 1$, which ensures the transition from a complex solution to two real solutions, is reached when the period is 140
days. The bottom panel shows the wavenumber as a function of the wave period computed from $W_R$ for 10° N (black), 15° N (grey) and 20° N (light grey). When $W_R > 1$, a condition which is always fulfilled at low frequency (characteristic time longer than 125 days at 15°), equation (16) has two real solutions. If $W_R \gg 1$ these solutions are close to $2W_R \simeq R_0\beta/\omega$ and $1/2W_R \simeq \omega/(\beta R_0)$. Consequently, the wave number $k$ is equal to either $\beta/\omega$ or $\omega/(\beta R_0^2)$. This result proves the existence of Rossby waves, whose wavelength is either short or long. When the frequency increases, $W_R$ decreases and eventually
reaches the critical value 1. When $W_R$ becomes smaller than 1, there are two complex conjugate solutions. The only acceptable solution has a *negative imaginary part* and conveys the existence of a Kelvin wave, trapped along the coastline and propagating northward. The absolute value of this imaginary part $k_I$ is represented as a dashed curve in Figure 10 (bottom panel). When the wave frequency is close to the critical value, $k_I$ vanishes and consequently the Kelvin wave is no longer trapped along the coast. The westward propagation of a Rossby wave with a significant amplitude can occur. At 15° N its wavelength will be
around 300 km. The zonal velocity (phase speed) of this non dispersive wave is about 2.5 km/day. On the other hand, as $l_R$ vanishes with $k_I$, the meridional velocity becomes infinite.

Similar results are obtained when the coast presents a constant angle $\theta$ with a meridian. The only change is that the critical frequency for which the wave regime changes is modified and depends on the tilt of the coast as indicated in Clarke and Shi (1991).

**Case of a cape.** The wave dynamics in a neighbourhood of the cape is characterized by the coefficients $W_R$ and $W_I$ which convey the effect of the coastline on the propagation of the wave. The coefficient $W_R$ can be splitted into two terms; the first one $\frac{R_0}{2b}\frac{2\xi^2 - 1}{\xi}\partial_Y\frac{F}{\omega}$ is similar to the term obtained from a straight coastline whereas the second one $\frac{R_0}{2a}\frac{1-\xi^2}{\xi}\partial_Y\frac{a}{b}$ explicits





the role of the coast. It is large when $\xi$ is small — in the frequency range where the model is valid, this means for wave periods going from $\sim 10$ days to a month — and when the deformation of the coastline is large. The existence of such a term was expected. When the angle of the coastline with a meridian increases, the impact of the latitudinal variations of the Coriolis parameter decreases along the path followed by the wave. It even vanishes when the coast becomes parallel to the equator.

These changes are taken into account by this term. On the contrary, at low frequency, the lengthening or shortening of the path followed by the wave becomes negligible because it occurs over a time which remains short in comparison with the period of the wave.

The variations of the coastline geometry also prevent the existence of two distinct solutions at low frequency. Indeed, $W_I$ differs from 0 and consequently two complex solutions are obtained as explicited in Figure 9. If $W_I$ is strictly negative (grey

area), the solution $z$ is inside the half unit disk and $k_R$ is small. If $W_I$ is strictly positive, the solution is outside and $k_R$ is large. The degeneracy of the equation thus desappears and a selection of the wavelength operates.

Figure 11 shows $W_R$ for $T = 10, 20, 50$, and $100$ days and makes visible an interesting property. When the period $T$ is equal to 10 days, a dissymmetry around the cape occurs: $W_R$ is negative south and positive north of the cape. This dissymmetry weakens when the period increases. For $T = 20$ days, the area where $W_R$ is negative is strongly reduced and for $T = 50$ days,

it has vanished. Since the signs of $W_R$ and $k_R$ are similar (see Figure 9 and the associated comments), a negative $k_R$ is expected in this area and actually appears (see Figure 13)

Since $W_I$ is nearly independent of the period in the considered frequency range, a single map suffices to describe it (Figure 12). $W_I$ is positive everywhere except in an narrow area west of the cape (the isoline -0.2 is indicated and the bold line corresponds to $W_I = 0$). Figure 9 shows that the corresponding values of $\xi z_1$ are smaller than 1. This suggests that this area is

occupied by waves whose wavelength is longer than everywhere else, a result which appears in Figure 13.

The maps of $k_R$ and $k_I$ (see Figures 13 and 14) show properties in agreement with the previous analysis. For periods shorter than 20 days, $k_R$ becomes negative south of the cape. Consequently the waves no longer propagate westward towards the open sea, but eastward towards the coast. On the contrary, north of the cape the propagation occurs always westward, whatever the frequency. At lower frequency this phenomenon is not observed. The coefficient $k_I$ shows a smaller dependence as a function

of the period. The shape of the Kelvin wave is modified close to the tip of the cape – its offshore extent is smaller since $k_I$ is larger – but it still exists and propagates northwards.

Lastly, note that the order of magnitude predicted in Figure 10 for a straight coastline with a south north orientation are noticeably changed when the shape of the coast is taken into account. For a wavelength of about 700 km, the corresponding period was equal to approximately 150 days. Now, figure 13 shows that this wavelength are obtained in a large part of the

domain for a period equal to 100 days. Note that this value is close to the values predicted by Clarke and Shi (1991) at the coast (between 84.69 and 115.2 days). Note also that, in a small area around the extremity of the cape (grey area), larger wavelengths are compatible with periods of about 100 days.





## 5   Conclusions

The analysis of satellite observations of the Chlorophyll showed a strong signal along and off the west african coast, between
10° and 22° N, in winter (December to April). Along the coast the high concentration of Chlorophyll is associated with the
offshore Ekman drift generated by the equatorward component of the trade wind, which forces an upward motion. Its variability
is modulated by Kelvin wave propagating northwards. In December 2002 and January 2003 we observed a wavelike pattern
in the open sea, which extends far away offshore, up to a distance of about 800 km off the coast. This signal was visible from
December 20$^{th}$ 2002 and was detectable during approximately one month, south of the Cape Verde Peninsula.

This pattern suggested the existence of locally generated Rossby waves, which slowly propagated westward. Indeed such
a wave can generate an elevation of the lower layers of the ocean corresponding to an upwelling of nutrient-rich water. We
investigated this possibility and the potential role of the cape, by first doing numerical experiments with a forced nonlinear
model, then by analytically studying a linear reduced gravity model.

The numerical study, based on a reduced gravity shallow water model, showed that a Rossby wave similar to the observed
pattern could be created by a wind burst broadly extending over the region where the oceanic signal was seen. Simultaneously,
a Kelvin wave formed along the coast and quickly propagated northwards. In our experiments a longshore wind burst which
lasted 5 days was used to generate the oceanic response. We showed that the spatial scale of the latter matches the spatial scale
of the forcing. The time scale of the reponse, controlled by the wavelength (see below), is not that of the forcing (5 days) but
much longer around 100 days for the first experiment.

The cape does not seem to modify the basic features of the wave dynamics. It mainly limits the extent of the wave to the
north. However, when the wind burst has a large zonal extent (of about 1000 km) and a small meridional extent (not exceeding
100 km), the response of the model close to the cape depends on the location of the wind anomaly. When the latter acts south of
the cape, the anomaly which forms around 15° W is small and quickly desappears; a wave which propagate southward seems
to prevent its existence. On the contrary, when the wind anomaly acts north of the cape, an anomaly forms and remains around
15° W, without moving; a secondary wave which propagates southward still exist but desappears quickly.

The analytical study, which extends the method suggested by Clarke and Shi (1991) to the open sea up to a distance of about
1000 km away from the coast, helps us interpret the numerical results and gives futher results. It first shows that a time scale
around 100 days can be associated with the observed wave, considering the value of the wavelength (around 700 km). This
value matches the critical value predicted by Clarke and Shi along the coast of Senegal, even though their model applied only
at the coast when the angle defining the tilt of the coastline is not too large. It also shows that the role of the cape does not
dramatically modify the dynamics of the system at such time scales.

On the contrary, when the period becomes shorter (smaller than 20-30 days), the waves behaves differently north and south
of the cape, as suggested by the numerical experiments. For the studied set-up, Rossby waves can propagate eastwards, in
a narrow band of the ocean whose latitudinal extent is about 100 km. We verified that this property vanished when the cape
flattened (the period of the wave progressively becoming shorter). This strongly suggests that the wave dynamics in the vicinity



of a cape – and the associated upwelling – depends on the geometry of the coastline for time scales shorter than one month. These changes no longer matter for longer time scales. Note that the behaviour difference predicted by the theory are not so important in the numerical experiments. This is not surprising since the geometry of the system is different in the numerical experiments.

Lastly this study suggests that offshore upwellings can be created or enhanced by Rossby waves. An example of such a phenomenom has been observed in the region off Senegal. This example is probably not unique. For instance Kounta et al. (2018) show patterns which provide clear evidence of an important Rossby wave activity to the south of the Cape Verde (see for example their Figure 10, which shows a climatology of the volume meridional transport). Observations will have to be pursued in this region and in other EBUS region to determine the importance of such events. However the observations of such structure

by satellite requires several condition which are seldom occuring together. First we need a long period of observations without clouds, second a typical wind event able to generate the Rossby wave and third the existence of nutrients in the subsurface layers, which could enrich the surface layers.

## Appendix A:  Appendix A

Since we consider processes whose time scale is larger than a day, equations (6) can be simplified. We define a characteristic

frequency $\omega_0$ and a daily frequency $F_d$ and assume that they verify $\epsilon = \omega_0/F_d \ll 1$. Setting $\omega_0 t = \tau$ and $F = F_d F_0$, the first two equations of system (6) become

$$
\begin{aligned}
\epsilon \partial_\tau T_X - F_0 T_Y &= R_X/F_d \\
\epsilon \partial_\tau T_Y + F_0 T_X &= R_Y/F_d
\end{aligned}
\tag{A1}
$$

with $R_X = -a^{-1}\partial_X \eta - b^{-1}\partial_Y \psi$ and $R_Y = -b^{-1}\partial_Y \eta + a^{-1}\partial_X \psi$. An elementary computation leads to the following relations

$$
\begin{aligned}
(\epsilon^2 \partial_{\tau\tau} + F_0^2) T_X &= (\epsilon \partial_\tau R_X + F_0 R_Y)/F_d \\
(\epsilon^2 \partial_{\tau\tau} + F_0^2) T_Y &= (\epsilon \partial_\tau R_Y - F_0 R_X)/F_d
\end{aligned}
\tag{A2}
$$

Considering our hypothesis, the terms of order 1 and $\epsilon$ can be kept in the previous equations and the terms of order $\epsilon^2$ can be neglected. Consequently we can use the approximate relations

$$
\begin{aligned}
T_X &= F^{-2}(\partial_t R_X + F R_Y) \\
T_Y &= F^{-2}(\partial_t R_Y - F R_X)
\end{aligned}
\tag{A3}
$$

(We used the fact that $\epsilon F_d^{-1}\partial_\tau = F^{-1}\partial_t$ and $F_0 F_d = F$). Note that the terms $\partial_t R_X$ and $\partial_t R_Y$ are of order $\epsilon$ in comparison with the terms $F R_Y$ and $F R_X$.

We now introduce these equations in the last equation of system (6). This leads to a new equation

$$
\partial_t \eta + \frac{C^2}{ab}[\partial_X(\frac{b}{F^2}(\partial_t R_X + F R_Y)) + \partial_Y(\frac{a}{F^2}(\partial_t R_Y - F R_X))] = \partial_t \phi
\tag{A4}
$$


or equivalently

$$\partial_t \left[ \eta - R_0^2 (\frac{1}{a^2}\partial_{XX}^2\eta + \frac{1}{b^2}\partial_{YY}^2\eta) \right] -$$
$$\frac{R_0^2}{ab}\left[ F^2\partial_X(\frac{b}{aF^2})\partial_{tX}^2\eta + \partial_Y F\partial_X\eta + F^2\partial_Y(\frac{a}{bF^2})\partial_{tY}^2\eta - \partial_X F\partial_Y\eta \right] = \partial_t\phi + R_\psi \qquad (A5)$$

where $R_0^2 = C^2/F^2$ is the Rossby radius and $R_\psi$ contains the forcing terms depending on $\psi$:

$$R_\psi = -\frac{C^2}{ab}\left[ \partial_X(\frac{1}{F^2})\partial_{tY}\psi + \partial_Y(\frac{1}{F^2})\partial_{tX}\psi + \partial_X(\frac{b}{aF}\partial_X\psi) - \partial_Y(\frac{a}{bF}\partial_Y\psi) \right]$$

The vanishing of the velocity orthogonal to the coordinates is easily obtained from system (A3). With the same approximations, the condition $T_X = 0$ at $X = 0$ implies $\partial_t R_X + F R_Y = 0$ or equivalently

$$b\partial_{tX}^2\eta + aF\partial_Y\eta = -a\partial_{tY}^2\psi + bF\partial_X\psi \qquad (A6)$$

for all $t > 0$ and $Y$.

When the forcing terms can be neglected, the previous relations simplify. They become respectively

$$\partial_t \left[ \eta - R_0^2(\frac{1}{a^2}\partial_{XX}^2\eta + \frac{1}{b^2}\partial_{YY}^2\eta) \right] - \cdots$$
$$\cdots \frac{R_0^2}{ab}\left[ F^2\partial_X(\frac{b}{aF^2})\partial_{tX}^2\eta + \partial_Y F\partial_X\eta + F^2\partial_Y(\frac{a}{bF^2})\partial_{tY}^2\eta - \partial_X F\partial_Y\eta \right] = 0 \qquad (A7)$$

and at $X = 0$, for all $t > 0$ and $Y$

$$b\partial_{tX}^2\eta + aF\partial_Y\eta = 0 \qquad (A8)$$

**Appendix B:  Appendix 2**

A method to solve system (13-14) with the boundary condition (15) is presented here. It has been shown that the values of $z_1$ and $z_2$ are known on the boundary thanks to equation (16). We set

$$z_1 = z_1^0 + \tilde{z}_1 \quad \text{and} \quad z_2 = z_2^0 + \tilde{z}_2$$

where $z_1^0$ and $z_2^0$ verifies equation (14) and the condition $z_2^0 = -i(\omega/F)z_1^0$ everywhere. Consequently, $z_1^0$ verifies equation (16)

everywhere

$$J(\xi z_1^0) = W_R + iW_I$$

and can be computed on the ocean domain. The results of this computation are presented in section 4.3 ($z_2^0$ is also known thanks to the relation $z_2^0 = -i(\omega/F)z_1^0$).



Since $z_1$ and $z_2$ are known on the coast and $z_1^0$ and $z_2^0$ everywhere, $\tilde{z}_1$ and $\tilde{z}_2$ are known on the coast $X = 0$. It is now easy to write the equations verified by $\tilde{z}_1$ and $\tilde{z}_2$.

$$\partial_Y(\frac{a\tilde{z}_1}{R_0}) - \partial_X(\frac{b\tilde{z}_2}{R_0}) = \partial_Y(\frac{az_1^0}{R_0}) - \partial_X(\frac{bz_2^0}{R_0}) = \mathcal{Z}_0 \tag{B1}$$

$$\tilde{z}_1^2 + 2\tilde{z}_1(z_1^0 - w_1) + \tilde{z}_2^2 + 2\tilde{z}_2(z_2^0 - w_2) = 0 \tag{B2}$$

with $\tilde{z}_1 = 0$ and $\tilde{z}_2 = 0$ for $X = 0$.

The variables $\tilde{z}_1$ and $\tilde{z}_2$ can be computed on a grid $(-i\Delta X, j\Delta Y)$ for $i = 0, 1, \ldots, N$, $j = -P, \ldots, -1, 0, 1, \ldots, P$. They are known for $X = 0$ ($i = 0$). We suppose that they have been computed for $i > 0$ (values $\tilde{z}_{1,i,j}$ and $\tilde{z}_{2,i,j}$) and show how $\tilde{z}_{1,i+1,j}$ and $\tilde{z}_{2,i+1,j}$ can be computed. Equation (B1) can be discretized in the following way:

$$\frac{b\tilde{z}_{2,i+1,j}}{R_0} = \frac{b\tilde{z}_{2,i,j}}{R_0} - \frac{\Delta X}{2\Delta Y}(\frac{a\tilde{z}_{1,i,j+1}}{R_0} - \frac{b\tilde{z}_{1,i,j-1}}{R_0}) + \Delta X \mathcal{Z}_{0,i,j}$$

the error being proportionnal to $\Delta X$. A boundary condition (for example $\tilde{z}_{2,i,-P} = 0$) is prescribed to end the computation. Knowing $\tilde{z}_{2,i+1,j}$, the value of $\tilde{z}_{1,i+1,j}$ is obtained by solving

$$\tilde{z}_{1,i+1,j}^2 + 2\tilde{z}_{1,i+1,j}(z_1^0 - w_1) + \tilde{z}_{2,i+1,j}^2 + 2\tilde{z}_{2,i+1,j}(z_2^0 - w_2) = 0 \tag{B3}$$

The computation of $z_1^0$ and $z_2^0$ would correspond to a solution such as $T_X = 0$ everywhere. The correction brought by $\tilde{z}_1$ and

$\tilde{z}_2$ is associated with a mass transport perpendicular to the coast. As the latter in general is much smaller than the transport $T_Y$ parallel to the coast, it is expected that $\tilde{z}_1$ and $\tilde{z}_2$ are small in comparison with $z_1^0$ and $z_2^0$ (they are null at $X = 0$ and increase proportionally to the distance to the coast). In section 4.3, the approximate value of $k$ associated with the solution $z_1^0$ is fully described.

*Author contributions.* MC provided the satellite analysis. JS made the numerical simulation. The manuscript benefitted from inputs of all

the authors

*Competing interests.* The authors declare that they have no conflict of interest

*Acknowledgements.* We thank X. Capet for fruitful discussions, for his careful reading of the manuscript and pertinent comments.



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



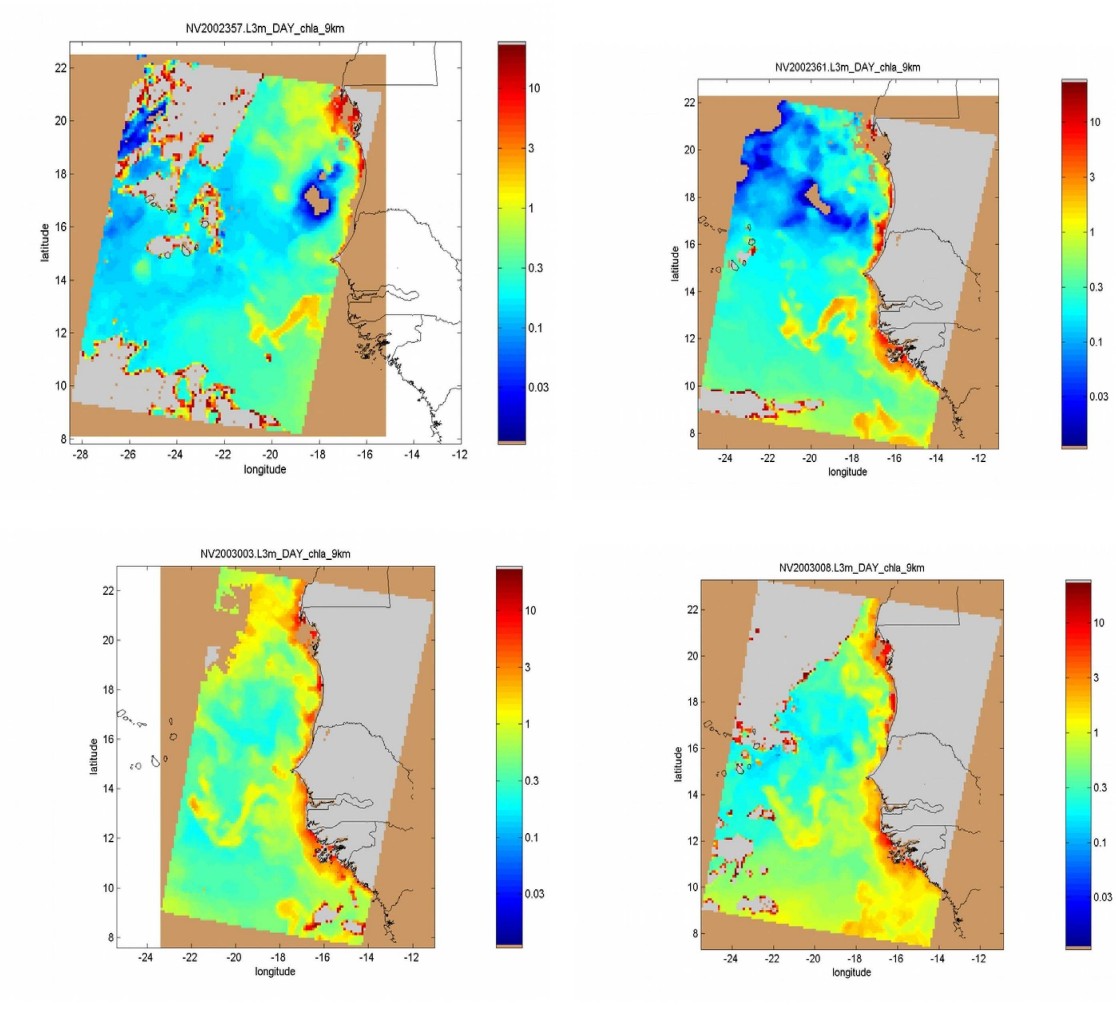

**Figure 1.** Chlorophyll observed at four different days (upper left: 23rd December 2002, upper right 27th December 2002, lower left: 3rd January 2003, lower right: 8rd January 2003). A wave-like pattern is clearly visible between $-22°$ and $-18°$ in longitude and $-12°$ and $-14°$ in latitude. At the end of the period a westward propagation seems to initiate.







**Figure 2.** Response of the ocean to an anomalous wind stress applied during five days (first panel) and corresponding evolution of the active layer thickness for every five days from the close of the the wind stress anomaly up to 35 days after (following panels). A Rossby wave is generated and a Kelvin wave quickly propagates along the coast. The Rossby waves very slowly propagates westward and its amplitude is divided by two between the first and the last panel. The 0 m isoline is indicated in bold; the isoline interval is 0.5 m (blue: negative).





**Figure 3.** Solution obtained for the same conditions as in Fig. 2 for a straight coasline. Note that the mean wind stress is added to the wind stress anomaly in the first panel. A Rossby wave is still generated but the halting of the signal due to the cape is no longer observed. The latitudinal extent of the wave east to 19°W is thus nearly twice larger than in the previous case.





**Figure 4.** In this experiment, the wind anomaly extent is four times smaller than the one used in Figure 2 and the position of the center is kept unchanged (first panel). It always acts during five days and as previously the active layer thickness is shown for every five days from the close of the the wind stress anomaly up to 35 days after. A Rossby wave is generated and its extent is approximately four times smaller than previously. No Kelvin waves are created because the wind stress anmomaly is nearly null along the coast. The 0 m isoline is indicated in bold; the isoline interval is 0.5 m (blue: negative).





**Figure 5.** The wind anomaly now acts over a long (1000 km) and narrow (200 km) domain south of the cape during five days (first panel). The solution after the close of the wind burst is shown. A wave is generated but the anomaly remains moderate near the coast. The 0 m isoline is indicated in bold; the isoline interval is 0.1 m (blue: negative).







**Figure 6.** The wind anomaly now acts over a long (1000 km) and narrow (200 km) domain south of the cape during five days (first panel). The solution after the close of the wind burst is shown. A wave is generated and the anomaly near the coast has an amplitude comparable with the anomaly around 25° W. The cape induces a dissymmetry of the response of the system though the forcing are identical. The 0 m isoline is indicated in bold; the isoline interval is 0.1 m (blue: negative).





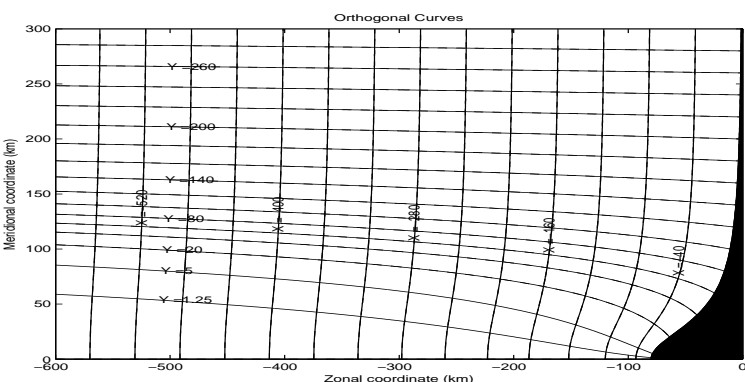

**Figure 7.** Example of orthogonal coordinates $\mathcal{X}$ and $\mathcal{Y}$ defined for a cape protruding from the coast into the sea over a distance of 80 km. Several values of $\mathcal{X}$ and $\mathcal{Y}$ have been indicated. Only one half of the domain has been represented.





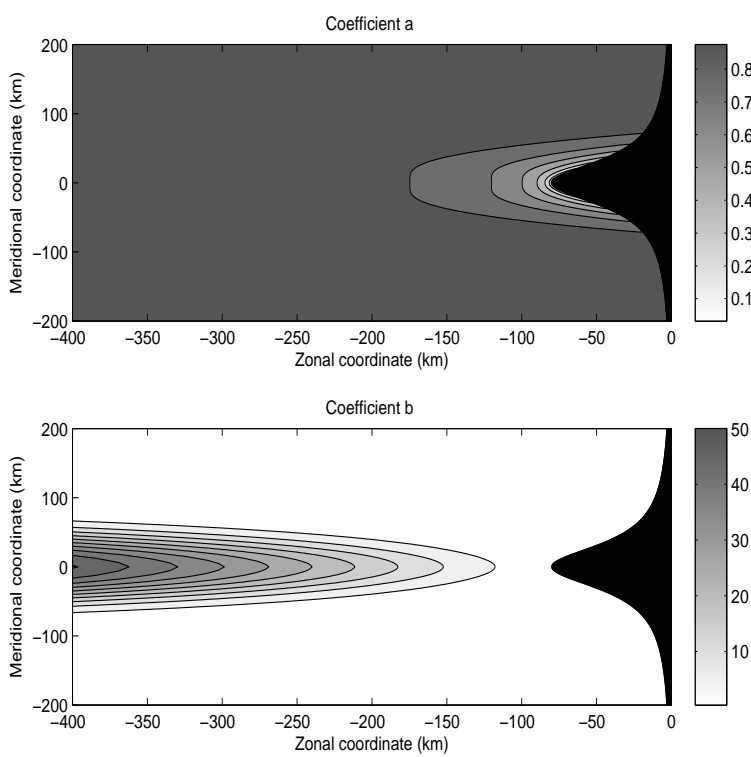

**Figure 8.** The coefficients $a$ and $b$ are given for the previous coordinate system (they have no dimension).



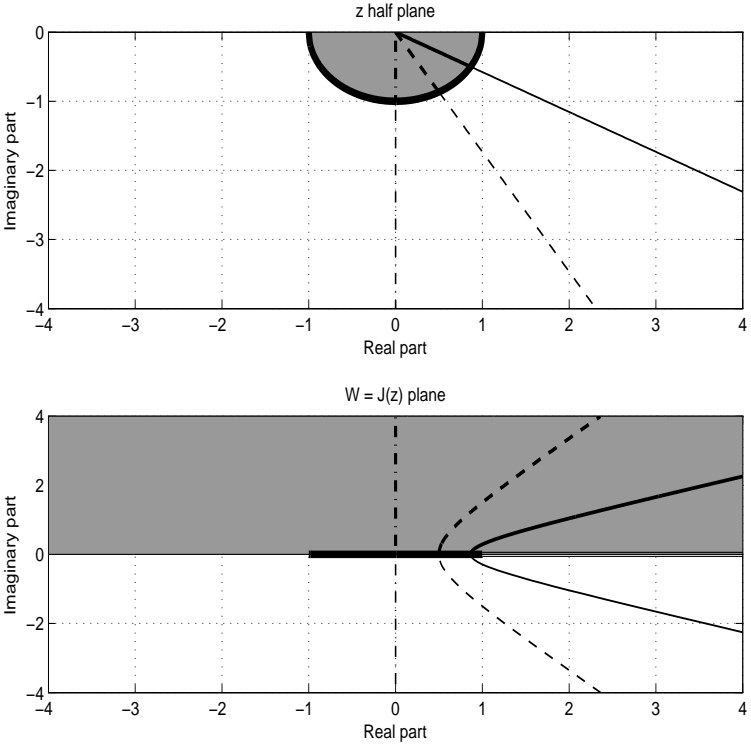

**Figure 9.** In top panel, the half complex plane $\Im(z) \leq 0$ with $z = \xi z_1$ is shown. The domain is restricted to complex numbers with a negative imaginary part since $k_I = a\Im(z)/(\xi R_0)$ must be negative. The half circle $|z| = 1$ and a few straight lines have been drawn. In botton panel the Joukowsky transform of the previous half plane $W = J(z)$ is shown. The segment $[-1, 1]$ is the image of the half circle. The upper (lower) half plane corresponds to the image of its interior (exterior). The image of the straight lines are correspondingly represented.





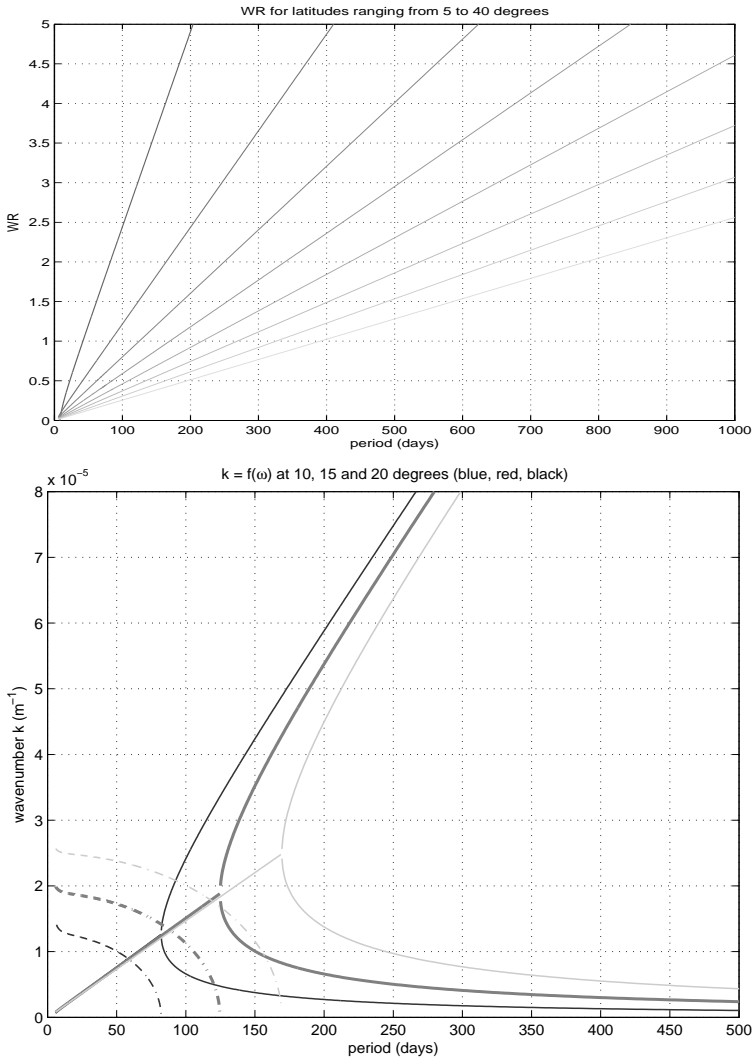

**Figure 10.** Top panel: $W_R$ is shown for latitudes ranging from $5°$ N (left segment) to $40°$ N (right segment). The critical value $W_R = 1$ is reached at $15°$ N for a period of about 125 days. Bottom panel: wavenumber $k$ as a function of the period $T$ for three different latitudes ($10°$ (black), $15°$ (grey thick curves), and $20°$). At high frequency, the waves are trapped along the coast since $k_I$ (dash-dotted curves) differs from 0 and Kelvin waves propagate along the coast. When the critical frequency, which decreases with the latitude, is reached, the trapped Kelvin waves no longer exist and Rossby waves can propagate. They have either a short or a long wavelength.





**Figure 11.** Dimensionless coefficient $W_R$ for periods equal to 10 days (left), 20 days (mid), 50 and 100 days (right). The bold black line corresponds to $W_R = 0$. When the period of the wave shortens (smaller than 20 days), $W_R$ becomes negative in a narrow band south of the cape. For period longer than 100 days, the diagram is nearly symmetric.



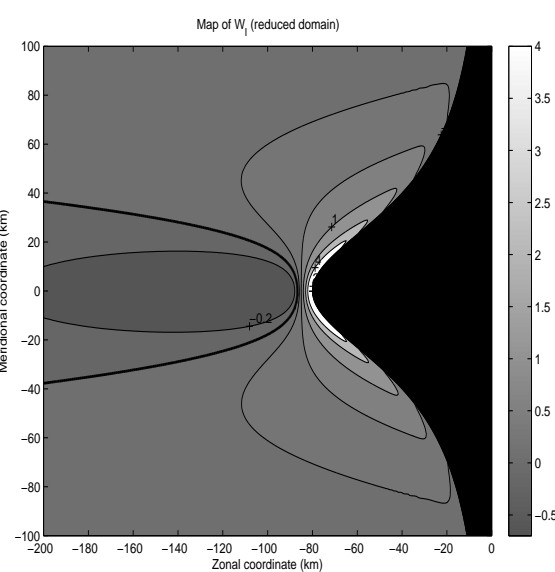

**Figure 12.** Dimensionless coefficient $W_I$. It is independent of the period of the wave. Note that the domain is reduced in comparison with the previous figure.





**Figure 13.** Coefficient $k_R$ for periods equal to 10 days (upper left), 20 days (upper right), 50 (lower left), and 100 days (lower right). The bold black line separates postive from negative values. Where $W_R$ is negative, $k_R$ is negative (for periods shorter than 20 days, in the area located to the south of the cape). This suggests the possibility of waves excited in the open sea with periods around a few days and trapped in the region located south of the cape.





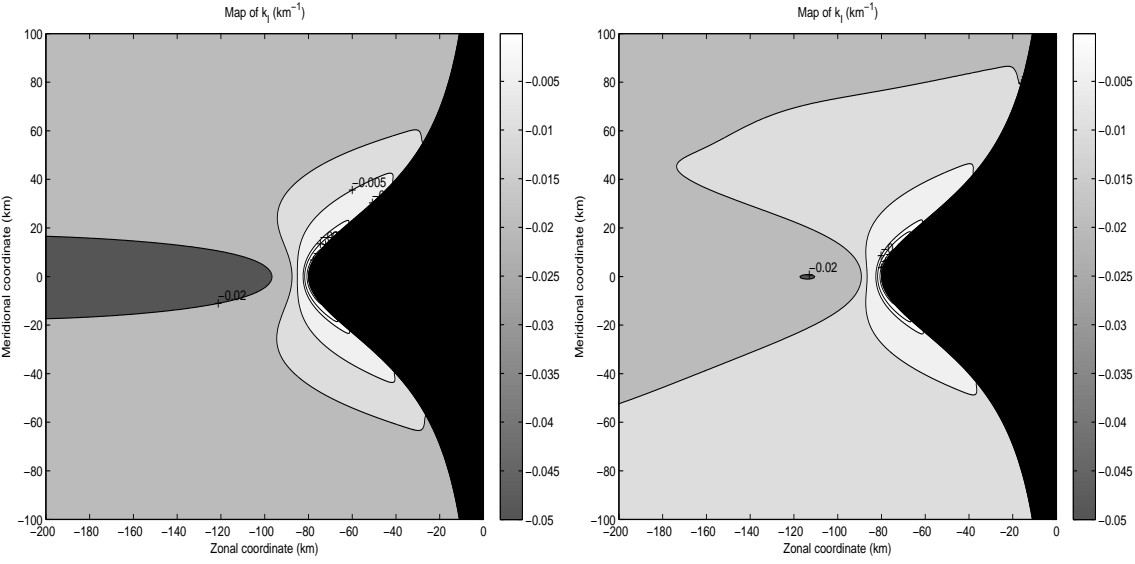

**Figure 14.** Coefficient $k_I$ for periods equal to 10 days (left) and 100 days (right). This coefficient weakly depends on the period of the wave.