# Peer review of "Generation of Rossby waves off the Cape Verde peninsula; role of the coastline"

_Ocean Science, 2019_

## Referee Comment (RC1) · Remi Tailleux (Referee) · 8 Jul 2019

**Summary and Recommendation** This paper explores the possibility to interpret some chlorophyll observations off Cape Verde in terms of a Rossby wave excited by a wind burst, which is supported by some idealised shallow-water numerical experiments. One full section is also dedicated to the theoretical analysis of the impact of the coastline on some of the waves properties such as the existence of a critical latitude. The result that a wind burst should excite both Kelvin and Rossby waves is expected from the existing literature and can hardly be regarded as new. The main novelty seems to be the extensive theoretical treatment of how the waves are impacted by the presence of a cape, which is addressed by rewriting the shallow-water equations in a general orthogonal curvilinear system of coordinates that has one of its coordinate align with

the coastline.

The combination of observations, idealised numerical modelling and theoretical analysis can potentially make the paper a strong one. The paper as currently written, however, feels a bit random at times, as it is not always clear why the authors do what they do, or why it is useful to do it. Moreover, the observational justification for the study is not really satisfactory, while the theoretical part is very hard to follow. I therefore recommend that the paper be returned for major revision aimed at improving the observational part and simplifying/streamlining the theoretical analysis before it can be considered for publication.

**Main comments**

1. The chlorophyll observations suggesting the presence of a Rossby wave are sketchy at best and hardly convincing, especially given the fact that they are only a very indirect proxy for dynamical activity, in contrast to sea surface height for instance. I believe that the paper would therefore be considerably enhanced by:

   - Adding Hoevmuller diagrams $SSH(x, y_i, t)$ at a number of selected latitudes $y_i(t)$ that could possibly provide some observational evidence of the critical latitude discussed by the authors using AVISO satellite altimeter data.

   - Adding hoevmuller diagrams of $SSH(x(s), y(s), t)$ where $(x(s), y(s))$ are points along the coastline that would indicate the presence of Kelvin or coastally-trapped waves, using AVISO satellite altimeter data. Several studies by Chris Hughes and Alban Lazar for instance have demonstrated that AVISO data can reveal the existence of such coastally trapped waves.

   - It would also be of interest to document the presence of wind bursts from a wind product

2. Numerical experiment. Can the authors justify the way they construct their reference state? How does the reference state impact on the various solutions with and without the cape? Is the solution very different from just imposing the wind burst anomaly on a resting state with a layer of uniform thickness?

3. The theoretical analysis is quite complex and hard to understand. The authors should make an effort as announcing upfront at the beginning of the section exactly what they aim to achieve, and what kind of questions they are trying to solve, and how this is supposed to inform the results of the numerical section. They should also provide more guidance to the reader as why they are doing what they are doing.

From a physical viewpoint, isn't it more usual to assume the alongshore flow to be geostrophic, i.e., to assume the following approximation

$$-FT_Y + a^{-1}\partial_X \eta = 0$$

$$\partial_t T_Y + FT_X + b^{-1}\partial_Y \eta = 0$$

$$\partial_t \eta + (C^2/(ab))[\partial_X(bT_X) + \partial_Y(aT_Y)] = 0.$$

It seems to me that this would lead to a simpler equation that the author's equation (7). Can the authors comment on this? Would we expect the results to be different?

As a result of the theoretical analysis, the authors arrive at some time scales, but it is not clear what these mean since they do not discuss how their shallow water model is calibrated, and to what extent it is representative of the characteristics of the region. Are these values sensitive to the choice of parameters?

**Other comments**

1. Abstract Line 5: "To verify this hypothesis" I would dispute that this hypothesis has been verified. To claim to have verfified this hypothesis, the authors need to expand the observational part of their paper to include Hoevmuller diagramms of SSH in the offshore and alonshore directions to check for th eexistence of westward propagating signals and coastally-trapped signals, preferanbly by correlating the with the presence of wind bursts.

2. Page 2, lines 7-9: "At a given frequency, there is a critical latitude at which the Kelvin wave no longer exists and are replaced by Rossby waves propagating westward" Probably need to add poleward of which Kelvin waves no longer exist.

3. page 3, Lines 7-8: "A well-defined "sline-like" pattern (figure 1): Could this be circled in the figure to make sure that there is no ambiguity for the reader as to what exactly the authors see in the figure?

4. Page 4, the model. Can the authors explain the usefulness of constructing a motionless background state that has a non-uniform layer thickness? How does that make the model more/less realistic? Can the authors demonstrate that this makes the stratification closer to the observed one in the region? How do the experiments differ if the perturbations are applied to a motionless constant thickness model instead?

5. Page 5, section 3.2. I think 'numerical resolution' should be 'numerical model setup' as I am not sure what 'resolution' mean here.

   • Can the authors explain how their model is calibrated exactly? Do they use one of the calibration methods proposed by Flierl (1981), and if so, which one? What are the climatological observations used for the calibration?
   • What are the boundary conditions used at the open boundaries?

6. Page 5, lines 27-28 "It thus suggest that the latter is a consequence of the existence of Rossby wave generated by a wind burst" THis is not implausible, as it is

difficult to think of what else this could be. Still, the authors should make an effort at significantly improving the observational basis for the hypothesis.

7. Page 6, lines 31: "comparable with the sine pattern observed" What is the basis for such a comparison? I don't really understand how a SSH signal can be compared with a chlorophyll signal, which are physiclally completely different kind of fields, one being passive, the other dynamical. Moreover, one should be proportional to $u' \cdot \nabla \overline{Chlorophyll}$, while the other is proportional to $h'$. Comparison would make more sense if westward propagation was seen in Aviso SSH data.

8. Page 8, Eq (7). Can the authors explain and justify the derivation of Eq. (7).

9. Page 9, Line 4, "When a and b are close to 1 [...]" This does not seem accurate, as for a pure rotation of the coordinates, which one would use do describe an inclined straight coastline, $a$ and $b$ would be exactly equal to unity, but $F$ would be a function of both $X$ and $Y$.

10. Page 9, Equation (10) In practice, this seems equivalent to assume the amplitude constant. How would you derive an equation for the amplitude otherwise?

11. Figure 2. Please indicate the timeline for each panel explicitly.

---

## Referee Comment (RC2) · Jacyra Soares (Referee) · 11 Jul 2019

General comments

The manuscript presents a verification study of the role of wind stress and of the coastline geometry in the generation of mesoscale anomalies offshore. For this purpose, it uses chlorophyll observations along the west African coast between 10° and 22° N, a reduced gravity shallow water model with a single active layer on the sphere and a theoretical analysis of the wave dynamics in the vicinity of a cover. The study is of interest for the scientific community. However, the generation of Rossby and Kelvin waves, at different frequencies, on the eastern ocean boundaries, considering the geometry of the coastline, has been studied for years. Therefore, I suggest that the authors perform

a major revision in the manuscript, indicating what results are numerically or theoretically new. The objective of the study needs to be properly highlighted and justified. Scientific findings and conclusions should also be presented more clearly.

Specific comments

Wave observation. I had difficulty seeing wave patterns and their basic characteristics (Figure 1) from the chlorophyll observations. This way, I suggest that plots of longitude (latitude) versus time, at particular latitude (longitude), would help viewing the zonal (meridional) propagation of Rossby and Kelvin waves, at a given latitude (longitude). The phase velocity of the waves would also be easily verified. If possible, I would like to see the observed mean wind field that generated the chlorophyll pattern shown in Figure 1.

Numerical model

- It is not clear, for me, if this is the first use of the numerical model described in the article. If it is not, please include reference of previous work. If it is the first use, detail how the calibration was performed; the model was able to reproduce properly the observations? How was this tested? Why did not you use a model already established by the scientific community?

- Please explain the value used for the dissipation coefficient and the effects that viscosity can have on baroclinic waves in shallow water numerical models.

- Boundary conditions: discuss the use of no-slip boundary condition. It is shown in the literature that the wave properties, particularly the longshore wave velocity, are much less dependent on the dissipation coefficient when the free-slip rather than no-slip conditions are used. At the open boundaries, what were the conditions used?

- The numerical model used in this work is simple and omits some potentially important factors which may influence poleward wave propagation, such as higher baroclinic modes, continental shelf topography and poleward variation of the thermocline. What

is the importance/impact of not considering such factors?

Analytical study and conclusion

- Section 4, "Analytical study", is hardly useful in its current form. I do not know if the way chosen to present the theoretical development along with the application of the theory is the most appropriate.

- Most of the theoretical development could be placed in the appendix. I suggest that only the essential expressions be left in section 4.

- The analytical study or its application is new?

- Please, in the "Conclusions" section emphasize novelty, what is different from others. Substantiate the novelty with comparison, analysis and/or applications.

Other comments

- Abstract, line 2; replace "kms" with km.

- Page 5, line 20-25: Please name the waves you are referring to.

- The current presentation of figures containing more than one panel (all figures except 7 and 12) is confusing. Please put an identification [(a), (b), etc or analogous] in the figure panels and describe it in the figure caption. I suggest that no result be discussed in the caption of the figures.

- Please, number within the text the coordinate system referenced in figure 8 and quote that number in the caption of the figure.

- Figure 10: It is difficult to understand and distinguish the lines and colors used.

- Figures 11 to 14: Very difficult to distinguish the values, especially the negative values of the positive ones.

---

## Author Comment (AC1) · 7 Oct 2019

Jérôme Sirven LOCEAN Université Pierre et Marie Curie T45 étage 4 CC100 4 place Jussieu 75252 PARIS CEDEX 05 email: js@locean-ipsl.upmc.fr

October 7, 2019

Dr R. Tailleux

Dear R. Tailleux,

you will find here our answer to the comments you made on our article "Generation of Rossby waves off the Cape Verde Peninsula; role of the coastline" which has been submitted to "Ocean science".

We first want to thank you. We apreciated your careful reading of the article and your numerous and interesting suggestions. We explain below how we took them into account. The corresponding modifications on the manuscript are also explicited.

1. You argued that the chlorophyll observations are only a very indirect proxy for dynamical activity and consequently must be completed by Sea Surface Height observations (for example coming from AVISO altimeter data) to convince the reader of the presence of Rossby waves. Consequently we have added a supplementary figure (new figure 2; see below) which is a Hoevmuller diagram of the SSH at different latitudes off the Cape Verde (12°N, 13.5°N, 15°N, and 16.5°N). This figure clearly shows the existence of a wave whose zonal velocity is about 4.5 cm/s, a value which corresponds to the estimation we have previously given from the observations and the theoretical study.

A figure of the wind stress has also been added (new figure 3; see below). This figure has been drawn from the ERAInterim reanalysis corresponding to the period December 2002-January 2003. The mean wind stress and the wind burst occuring around December 10th are both shown.

Note that we decided not to add a supplementary figure to document the existence of Kelvin waves propagating along the coast but only to add references about the work of A. Lazar in introduction and conclusion. Here are the changes in the introduction

Attention is focused on offshore mesoscale activity associated with the upwelling, a recurrent feature of upwelling systems (see Capet et al., 2008 a, b). The alongshore activity, which has received much more attention (see for example Ndoye et al., 2016 and the references herein) is not studied. and in the conclusion.

The existence of a Chlorophyll signal far from the coast – here extending up to 750 km west to the Cape Verde has to our knowledge never been described. This strongly differs from the coastal signals associated with Kelvin waves which have been previously carefully analyzed (see Ndoye et al. 2016 and the references herein).

Indeed we think that an originality of this paper is to consider signals – and specifically biological signals – far away from the coast. The dynamics of the Kelvin waves, which remains trapped to the coast, have received in this region much more attention and the works of A. Lazar exemplifies this. We thus prefered to emphasize the existence of the Rossby wave and its biological signature.

Just below is given in italic the text which has been introduced about AVISO data on page 4 The corresponding figure is shown at the end of this text.

... To corroborate this hypothesis, we analyzed the Sea Surface Height (hereafter SSH) obtained from AVISO satellite altimeter data for the corresponding period (December 2002 - January 2003). Hoevmuller diagrams are shown at  $12^{\circ}$ ,  $13.5^{\circ}$ ,  $15^{\circ}$ , and  $16.5^{\circ}$  in figure 1; they clearly confirm the existence of a Rossby wave propagating westwards with a velocity of about 4.5 cm s-1. The amplitude of this wave becomes smaller northwards: it peaked at 13 cm between  $12^{\circ}$  and  $13.5^{\circ}$  degrees but did not exceed 7 cm at  $16.5^{\circ}$ . The wavelength is around 700 km, comparable with the extent of the Chlorophyll signal.

2. Numerical experiments. We have added a new paragraph to respond to your queries. We have chosen a mean state which more or less corresponds to the mean state observed in this region (a dominant wind stress blowing from the north-north-east (see the new figure 3) but which remains simple enough to permit the ulterior analytical work. The reference state has a negligible impact on the solution – only some details are modified – and we decided not to develop this point. However, we had to verify that our results subsist in a more complex (and more realistic) situation than the one where the state is at rest with a unifrom layer. The model has been calibrated so that the Rossby radius has a reasonable value in this area and consequently that the propagation velocity of the waves is realistic. Actually, the phase velocity of the waves is about 4. cm s-1, which corresponds to the phase velocity of the waves shown by AVISO data.

Just below is given in italic the text of the subsection which has been added in section 2 and the corresponding figure.

**...3.3 Numerical set-up of the model**

In Figure 2 the mean wind for the considered period (December 2002-January 2003) is shown. It exemplifies the situation which is normally found in this

region. The wind regularly blows from the north-north-east with a velocity ranging from 4 to 8 m s-1. To take this into account, a constant mean wind stress of amplitude equal to  $0.06 \text{ N m}^{-2}$  (corresponding to a mean wind velocity of about 5 m s-1 and a value of  $\tau_0$  equal to  $6 \times 10^{-5} \text{ m}^2 \text{ s}^{-2}$ ) and oriented along a south-south-west direction is applied from rest during four years, until a stationary mean state, which verifies the theoretical relation given in section 3.1, is reached.

As shown in Figure 2 a wind anomaly was active when the wave of Figure 1 begins to be observed. This anomaly obviously is transient, but to the south of the Cape Verde, it mainly points southwards. To represent this situation in a simplified way, we defined in a first experiment a north-south wind stress anomaly which extends over approximately 500 km and whose maximum is still equal to 0.06 N m-2. This anomaly is applied during five days (see Fig. 4, first panel). The integration is continued during 45 days, after the anomaly has disappeared.

To explore the sensibility of the model response to the wind anomaly, others wind anomalies have been applied (see below, in particular Figures 6 to 8). The results obtained for these anomalies are discussed in the next section.

3. You judged that the theoretical analysis is hard to understand. We followed your advice and thus summarized as explained below the major lines of the analysis to guide the reader. We think that a reader who is not interested in the details of the mathematics can now skip some computations.

In this section, we aim at understanding if the coastline may influence the propagation of the Rossby waves which are created close to the coast and propagate towards the open sea. The impact of the coastline has been investigated by Crépon and Richez (1984), Clarke (1977), and Clarke and Shi (1991) for the Kelvin waves using an analytical approach. Here we focus on the Rossby waves; as we consider an area which extends up to about 1000 km from the coast, we have to generalize the approach followed by Clarke and Shi, which introduced a local system of coordinates dependent of the coastline to study Kelvin waves along an irregular coastline. We try to answer the following questions:

a) Are there time scales for which the impact of the coastline (small in the numerical experiments) becomes more important ?

b) A dissymetry between the response north and south of the Cape was visible in the numerical experiments; can this dissymmetry be dependent on the existence of the Cape ?

The analysis begins by defining and building a system of coordinates that permits to follow the coastline geometry. This procedure is a standard one in mathematics when boundaries are complex; indeed the the boundary conditions can be simply written, which constitutes a substantial advantage. however, it has a drawback: the differential equations which characerize the problem become slightly more complex because they must include geometrical factors that take into account the deformations associated with the new system of coordinates. This drawback is small in comparison with the advantage.

When these new equations are established, staightforward calculations are made to obtain a unique partial differential equation (equation (7)), which characterizes the evolution of  $\eta$  (the thickness of the active layer). This equation is a wave equation. Consequently the ray theory (or equivalently the WKB method) can be applied. When the forcing terms are neglected, this yields a first order nonlinear differential equation (equation (11)). No new ideas are introduced after this. We just rewrite equation (11) by introducing new notations, in order to facilitate its study and the presentation of the results (end of paragraph 4.2). We then describe the results when the tranport along the coast is much larger than the transverse transport (section 4.3).

We also added the following lines after the system of equations (7)

Note that we use a more complex system than the traditional one which is used for the study of the Kelvin waves and which neglect the variations of the transport perpendicular to the coast :

$$\begin{cases} -FT_Y + a^{-1}\partial_X \eta &= -b^{-1}\partial_Y \psi \\ \partial_t T_Y + FT_X + b^{-1}\partial_Y \eta &= a^{-1}\partial_X \psi \\ \partial_t \eta + (C^2/(ab))[\partial_X(bT_X) + \partial_Y(aT_Y)] &= \partial_t \phi \end{cases}$$

We use the complete system (7) because we study Rossby waves far from the coast, for which these hypotheses does not hold. Indeed, the previous simplified system leads to drop in equation (11) below two terms:  $(1/b^2)(\partial_Y \theta)^2$  and  $iF^2 \partial_Y(a/(bF^2)) \partial_Y \theta$ . These terms include the geometrical factors a and b due to the coordinates change; and we precisely try to investigate which is the impact of such terms.

Lastly, note that the time scale we obtain are only dependent of the speed C for a given geometric configuration. The value of C depends on the mean state and on the initial thickness of the model. A larger C leads to shorter time scales.

4. I answer more briefly to the eleven other comments since many of them are strongly linked to the previous major points.

Abstract line 5. As previously explained, we have now introduced Hoevmuller diagrams from AVISO altimeter data to show that direct observations of the ocean showed the existence of a westward propagating signal. The Abstract has been modified in consequence.

**Page 2, lines 7-9** We added as required "poleward of which Kelvin waves no longer exist".

Lines 7-8 The well defined "sine like" pattern has been circled as required.

**Page 4; the model** This point is close to the second point of the main three comments. We constructed a background state which has a non uniform layer because this non uniform layer could have an impact on the Rossby waves propagation (like the  $\beta$  topographic effect). It is slightly more realistic than a uniform layer since the isopycns are not flat in this area. We must avow however that the experiments do not differ much from experiments made with a uniform layer.

**Page 5, section 3.2** We introduced as suggested a new subsection named "numerical model setup" and limited the subsection "numerical resolution" to technical points. We do not use one of the calibration methods proposed by Flierl (1981). The climatological observations we used was for the wind stress the climatology of ERA Interim – as explained above a picture of the mean wind in December, 2002 is now shown.

At the open boundaries, we simply assumed that the velocity vanished.

We want to emphasize that the aim of the paper is not to reproduce precisely the climatological characterisitics of the region (the paper of Kanta et al., 2018 shows how a GCM can successfully reproduce the complexity of this region) but to propose a simple model which could explain some observations seen off the Cape Verde.

**Page 5, Lines 27-28** As explained above, we tried to improve significantly the observational basis justifying this hypothesis by introducing AVISO altimeter data and adding a figure of the wind stress computed by ERAInterim.

**Page 6, line 31** As you suggested, we now have introduced Hoevmuller diagram of the sea surface height which clearly shows the propagation of a wave with a velocity of  $4.5 \text{ cm}s^{-1}$ . More details can be found at the beginning, in our answer to the first comment.

**Page 8** The equation is completely justified in Appendix A. The idea is very simple:

$$\partial_t T_X - F T_Y = -a^{-1} \partial_X \eta \Longrightarrow \partial_{tt}^2 T_X - F \partial_t T_Y = -a^{-1} \partial_{tX}^2 \eta$$

Using the second equation we deduce

$$\partial_{tt}^2 T_X + F^2 T_X = -a^{-1} \partial_{tX}^2 \eta - F b^{-1} \partial_Y \eta$$

Doing similar computations from the second equation, we obtain an equation for  $T_Y$  of the same type:

$$\partial_{tt}^2 T_Y + F^2 T_Y = -b^{-1} \partial_{tY}^2 \eta + F a^{-1} \partial_X \eta$$

We now have just to apply the operator

$$\partial_{tt}^2 + F^2$$

to eliminate  $T_X$  and  $T_Y$  and obtain an equation for  $\eta$  only. In appendix A the computation are done, taking into account our hypothises on the time scale of the motion.

**Page 9, line 4** We have modified the sentence as follows When a and b are close to 1 and the coast has a south north orientation (see Figure 8) the new coordinates system is nearly similar to the original one; indeed F mainly depends on  $Y \simeq y$  only and in those regions equation (7) thus simplifies:

$$\partial_t \left[ \eta - R_0^2 (\partial_{XX}^2 \eta + \partial_{YY}^2 \eta) \right] - R_0^2 \left[ \partial_Y F \, \partial_X \eta + F^2 \partial_Y (\frac{1}{F^2}) \, \partial_{tY}^2 \eta \right] = 0 \quad (1)$$

We recognize the equation characterizing the propagation of waves in the  $\beta$ plane ( $\beta = \partial_Y F$ ) for a shallow water model. For a tilted coast the dependance of the Coriolis parameter as a function of X should be still taken into account.

**Page 9, Equation (10)** We have not developped this point in the paper but an equation for  $\eta_0(X, Y)$  can be established and allows to predict the evolution of the amplitude. It is obtained by putting  $\eta$  in equation (7) and writing all the resulting terms; this leads to an equation which contains  $\eta_0$  and  $\theta$ . The hypotheses we made leads to solve the equation for  $\theta$  which is given (equation (12)) and which contains information about the wave propagation. Once  $\theta$  is known,  $\eta_0(X, Y)$  can be determined using the complete equation. Generally, it is not very interesting except if critical latitude exist (in geometrical optics: focus). The amplitude of the wave increases and in the most simple cases can be represented using Airy functions.

Figure 2. The timeline for each panel has been indicated explicitly.

Sincerely yours,

Jérôme Sirven

The new figures are just below.

---

## Author Comment (AC2) · 7 Oct 2019

Jérôme Sirven LOCEAN Université Pierre et Marie Curie T45 étage 4 CC100 4 place Jussieu 75252 PARIS CEDEX 05 email: js@locean-ipsl.upmc.fr

October 7, 2019

Dr J. Soares

Dear J. Soares,

you will find here our answer to the comments you made on our article "Generation of Rossby waves off the Cape Verde Peninsula; role of the coastline" which has been submitted to "Ocean science".

We first want to thank you. We appreciated your careful reading of the article and your numerous and interesting suggestions. We explain below how we took them into account. The corresponding modifications on the manuscript are also explicited.

You first suggest us to indicate more precisely which is new in the paper. We modified the conclusion to emphasize the novelties. Three points seemed important to us :

1. The existence of a Chlorophyll signal far from the coast – here extending up to 750 km west to the Cape Verde has to our knowledge never been described. This strongly differs from the most common signals which ar found along the coast and which can be easily associated with coastal Kelvin waves.

2. The method we developped to decipher the mechanisms at play in this region is new. It allows us to study up to 1000 km from the coast the wave dynamics. This method generalizes the method previously used by Clarke (1980).

3. Combining this analytical work, observations of SSH and numerical experiments we have been able to associate the existence of this signal to the propagation of a Rossby wave generated by a wind burst. We showed that the duration of the wind burst has not a direct relation with the period of the waves and that the size of the wind burst was more relevant. Lastly, we proved that the pattern of the coast does not not have in this case a prominent role because the wave has a long period (longer than two months). We showed that a strong impact could be only expected if the wave period was shorter than about 15 days.

In italic are the main changes we have done in the conclusion:

This pattern suggested the existence of locally generated Rossby waves, which slowly propagated westward. Indeed such a wave can generate an elevation of the lower layers of the ocean corresponding to an upwelling of nutrient-rich water. The existence of this wave was confirmed by the study of the SSH signal coming from AVISO satellite data. It evidenced a wave propagating westward with a velocity of about 4.5 cm s-1. The existence of a Chlorophyll signal far from the coast – here extending up to 750 km west to the Cape Verde – has to our knowledge never been described. This strongly differs from the coastal signals associated with Kelvin waves which have been previously carefully analyzed (see Ndoye et al. 2016 and the references herein).

In this study we thus investigated the mechanisms which could lead to the existence of such a wave and analyzed the potential role of the cape, by first doing numerical experiments with a forced nonlinear model, then by analytically studying a linear reduced gravity model.

The analytical study is new and extends the method suggested by Clarke and Shi (1991) to the open sea up to a distance of about 1000 km away from the coast. It helps us interpret the numerical results and gives further results.

An important problem is the detectability of these waves. Dealing with a reduced gravity model whose characteristics are fitting the observations, we found that the elevation of the interface probably does not exceed a few meters. The interface elevation facilitates the nutrient enrichment of the surface layers and consequently favours phytoplankton blooms. As the elevation of the interface is relatively small, the phytoplankton bloom is likely to occur only under very specific conditions such as a relatively small average thermocline depth or the presence of phytoplankton species capable of rapid growth, with a strong chlorophyll signature like diatoms. In fact, phytoplankton pigment retrieval from ocean color satellite observation shows that the chlorophyll signal we observed is dominated by fucoxanthin, which is a signature of diatoms (Khalil et al, 2019, submitted)

We are concious that this study is preliminary. In particular we think that it would be of interest to look for Chlorophyll signals up to 1000 km off the coast since satellite data have been available for twenty years. The existence of such blooms far from the coast could be of interest for the economy of fishery in Senegal and Gambie.

Then you make more specifc suggestions, which are generally very close to those made by the first reviewer. You suggest us to document more precisely from other observations than Chlorophyll the signal we observed, to give more details about the numerical model, to improve the presentation of the analytical study, and to modify the conclusion to put in light more clearly the novelty of the results (this corresponds to the general comment about the paper and we have just given our response above). Six other comments were made to improve the lisibility of the figure and text. I detail below how these comments have been addressed.

1. Wave observations To make more visible the Chlorophyll pattern we consider, we have cercled it. As you suggest, we have completed this observation by Sea Surface Height observations coming from AVISO altimeter data. Consequently we have added a supplementary figure (new figure 2) which is a Hoevmuller diagram of the SSH at different latitudes off the Cape Verde (11°N, 13.5°N, 15°N, and 16.5°N). This figure clearly shows the existence of a wave whose zonal velocity is about 4.5 cm/s, a value which corresponds to the estimation we have previously obtained from the observations and the theoretical study.

Here is the text we have added:

... To corroborate this hypothesis, we analyzed the Sea Surface Height (hereafter SSH) obtained from AVISO satellite altimeter data for the corresponding period (December 2002 - January 2003). Hoevmuller diagrams are shown at  $12^{\circ}$ ,  $13.5^{\circ}$ ,  $15^{\circ}$ , and  $16.5^{\circ}$  in figure 1; they clearly confirms the existence of a Rossby wave propagating westwards with a velocity of about 4.5 cm s-1. The amplitude of this wave becomes smaller northwards: it peaked at 13 cm between  $12^{\circ}$  and  $13.5^{\circ}$  degrees but did not exceed 7 cm at  $16.5^{\circ}$ . The wavelength is around 700 km, comparable with the extent of the Chlorophyll signal.

A figure of the wind stress has also been added (new Figure 3). This figure has been drawn from the ERAInterim reanalysis corresponding to the period December 2002-January 2003. The mean wind stress and the wind burst occuring around December 8 are both shown. This figure has been introduced in the section about the numerical model (see just below) to justify how we calibrate it.

Note that we decided not to add figures to document the existence of Kelvin waves propagating along the coast but only to add new references. Indeed we think that an originality of this paper is to consider signals – and specifically biological signals – far away from the coast. The dynamics of the Kelvin waves in this area have received much more attention (the work of A. Lazar exemplifies this fact) and consequently we prefered to emphasize the existence of the Rossby wave and its biological signature.

**2. The model**

First of all this model has been used in different configurations (with one or two active layers). References can be found for example in the following publication: Février, S., J. Sirven, C. Herbaut, 2007: Interaction of a coastal Kelvin wave with the mean state in the gulf Stream separation area, *J. Phys. Oceanography*, which has been added. Note that the numerical technique that is used (enstrophy conserving) is the same as the one used in the OGCM NEMO (which is quoted in much more than one hundred articles; see for example the publications with G. Madec as coauthor).

We made several test on the dissipation coefficient. It damps the response of the model as expected from theory. In the retained configuration, the amplitude of the anomaly is divided by 2 after 45 days, which corresponds to the damping coefficient we used. We chose to use a resolution of  $1/12^{\circ}$  (much smaller than the Rossby radius of deformation  $\simeq 55$  km: eddy permitting

model) and a small viscosity coefficient. Surprisingly, when we made experiments at a coarser resolution, the results were not dramatically modified.

As indicated in the text, no slip boundary conditions are used everywhere. We are concerned here by the offshore velocity of the Rossby waves which weakly depends on the dissipation coefficient and the boundary conditions. A discussion of Kelvin waves velocity in simple models can be found in Février et al. 2007.

We have chosen a mean state which more or less corresponds to the mean state observed in this region (a dominant wind stress blowing from the north-north-east (see the new figure 3) but which remains simple enough to permit ulterior analytical work. The reference state has a negligible impact on the solution – only some details are modified and we decided not to develop this point. However, we had to verify this to be sure that our results were not dependent of a simpler choice – for example a resting state with a uniform layer which does not take into account the poleward variation of the topography. The model has been calibrated so that the Rossby radius has a reasonable value in this area and consequently that the propagation velocity of the waves is realistic. Actually, the phase velocity of the waves is about 4. cm s-1, which corresponds to the phase velocity of the waves shown by AVISO data.

Note that the paper is devoted to the study of Rossby waves which mainly propagate westwards. The model we use – which would be perhaps too simple to study the poleward propagation of coastal Kelvin waves – seems ro reproduce, albeit its simplicity, the westward propagation of Rossby waves that have left the continental shelf.

Just below is given in italic the text of the subsection which has been added in section 2 and the corresponding figure showing the observed mean wind stress and the wind anomaly.

**...3.3 Numerical set-up of the model**

In Figure 2 the mean wind for the considered period (December 2002-January 2003) is shown. It exemplifies the situation which is normally found in this region. The wind regularly blows from the north-north-east with a velocity ranging from 4 to 8 m s-1. To take this into account, a constant mean wind stress of amplitude equal to  $0.06 \text{ N m}^{-2}$  (corresponding to a mean wind velocity of about 5 m s-1 and a value of  $\tau_0$  equal to  $6 \times 10^{-5} \text{ m}^2 \text{ s}^{-2}$ ) and oriented along a south-south-west direction is applied from rest during four years, until a stationary mean state, which verifies the theoretical relation given in section 3.1, is reached.

As shown by Figure 2, a wind anomaly was active when the wave of Figure 1 begins to be observed. This anomaly obviously is transient, but to the south of the Cape Verde, it mainly points southwards. To represent this situation in a simplified way, we defined in a first experiment a north-south wind stress

anomaly which extends over approximately 500 km and whose maximum is still equal to 0.06 N m-2. This anomaly is applied during five days (see Fig. 4 first panel). The integration is continued during 45 days, after the anomaly has disappeared.

To explore the sensibility of the model response to the wind anomaly, others wind anomalies were applied (see below, in particular Figures 6 to 8). The results obtained for these anomalies are discussed in the next section.

**3. Analytical study**

Clearly this section needed to be improved. We have put a maximum of expressions in Appendix A to alleviate the sufferings of the reader but found difficult to shift section 4.2 from the main text to a new Appendix. Consequently, we decided to follow the suggestion of Rémi Tailleux (first reviewer) and summarized at the beginning of the analytical study the main steps of the study.

Here is the text which has been introduced in section 4.

In this section, we aim at understanding if the coastline may influence the propagation of the Rossby waves which are created close to the coast and propagate towards the open sea. The impact of the coastline has been investigated by Crépon and Richez (1984), Clarke (1977), and Clarke and Shi (1991) for the Kelvin waves using an analytical approach. Here we focus on the Rossby waves; as we consider an area which extends up to about 1000 km from the coast, we have to generalize the approach followed by Clarke and Shi, which introduced a local system of coordinates dependent of the coastline to study Kelvin waves along an irregular coastline. We try to answer the following questions:

a) Are there time scales for which the impact of the coastline (small in the numerical experiments) becomes more important ?

b) A dissymetry between the response north and south of the Cape was visible in the numerical experiments; can this dissymmetry be dependent on the existence of the Cape ?

The analysis begins by defining and building a system of coordinates that permits to follow the coastline geometry. This procedure is a standard one in mathematics when boundaries are complex; indeed the the boundary conditions can be simply written, which constitutes a substantial advantage. however, it has a drawback: the differential equations which characerize the problem become slightly more complex because they must include geometrical factors that take into account the deformations associated with the new system of coordinates. This drawback is small in comparison with the advantage.

When these new equations are established, staightforward calculations are made to obtain a unique partial differential equation (equation (7)), which characterizes the evolution of  $\eta$  (the thickness of the active layer). This equation is a wave equation. Consequently the ray theory (or equivalently the WKB method) can be applied. When the forcing terms are neglected, this yields a first order nonlinear differential equation (equation (11). No new ideas are introduced after this. We just rewrite equation (11) by introducing new notations, in order to facilitate its study and the presentation of the results (end of paragraph 4.2). We then describe the results when the tranport along the coast is much larger than the transverse transport (section 4.3).

Yes, the analytical study is new (as well as the numerical experiment, even though the model has been already uesd).

- 4. **Conclusion** We have already considered this point at the beginning of our answer.
- 5. Other comments

Abstract line 2 we replaced "kms" with "km".

Page 5, line 20-25 of the previous manuscript. The wave we are referring to have been named.

**Figures** The presentation of the Figures have been modified. We have put an identification in the figure panels and described the panel in the figure caption as you suggested. No results are discussed in the caption of the Figures.

The coordinate system referenced in Figure 8 is now explicitly named to avoid any confusion.

Figure 10 has been redrawn (in color) to increase its lisibility.

Figures 11 to 14 have been redrawn (in color) to increase their lisibility.

Sincerely yours, Jérôme Sirven

The new figures are just below.